## Registered report

behaviour/psychology

free will, determinism, beliefs, attitudes, experimental philosophy, open science

**Author for correspondence:**
David Wisniewski
e-mail: david.wisniewski@ugent.be

# Cultural pressure and biased responding in free will attitudes

Emiel Cracco[1,†], Carlos González-García[1,†], Ian Hussey[2], Senne Braem[1,3] and David Wisniewski[1]

[1]Department of Experimental Psychology, and [2]Department of Experimental Clinical and Health Psychology, Universiteit Gent, Ghent, Belgium
[3]Department of Psychology, Vrije Universiteit Brussels, Brussels, Belgium

 EC, 0000-0003-4043-5992; CG-G, 0000-0001-6627-5777;
IH, 0000-0001-8906-7559; SB, 0000-0002-2619-8225;
DW, 0000-0003-4793-1171

Whether you believe free will exists has profound effects on your behaviour, across different levels of processing, from simple motor action to social cognition. It is therefore important to understand which specific lay theories are held in the general public and why. Past research largely focused on investigating free will beliefs (FWB, 'Do you think free will exists?'), but largely ignored a second key aspect: free will attitudes (FWA, 'Do you like/value will?'). Attitudes are often independently predictive of behaviour, relative to beliefs, yet we currently know very little about FWAs in the general public. One key issue is whether such attitudes are subject to biased, socially desirable responding. The vast majority of the general public strongly believes in the existence of free will, which might create cultural pressure to value free will positively as well. In this registered report, we used a very large ($N = 1100$), open available dataset measuring implicit and explicit attitudes towards free will and determinism to address this issue. Our results indicate that both explicit and implicit attitudes towards free will are more positive than attitudes towards determinism. We also show that people experience cultural pressure to value free will, and to devalue determinism. Yet, we found no strong evidence that this cultural pressure affected either implicit or explicit attitudes in this dataset.

## 1. Introduction

Whether we have free will or not is a question that has been debated among philosophers for centuries. This academic debate largely revolved around the question of whether free will can exist in a world that is deterministic [1–3]. More recently, research became interested in mapping out lay theories of free will, asking the

†These authors contributed equally to this study.

general public whether free will exists [4]. These findings indicate that the vast majority of the general public believes in free will [5]. Interestingly, this free will belief (FWB) is not unshakable, and can be reduced experimentally [6]. There is some evidence that the weakening FWBs could lead to adverse effects such as antisocial behaviour [6–8], but see [9]. In recent years, it has been demonstrated how our behaviour is determined by unconscious neural processes [10–12]. Such findings might reduce FWBs, and already led some to conclude that free will is an 'illusion' [13]. This claim potentially has broad effects, as free will is the foundation for our sense of responsibility and many criminal justice systems [14–16]. If FWBs changed in the general public, this might lead to unintuitive changes to e.g. our sense of responsibility [17]. Importantly, whether and how our FWBs might change based on conflicting information probably depends on the attitude we have towards free will (FWA): people with a positive FWA will be more reluctant to change their beliefs than people with negative FWAs. Therefore, this study presents a comprehensive analysis of FWAs in the general public, which is currently missing in the literature.

## 1.1. Free will beliefs versus free will attitudes

In the free will belief literature, FWBs are predominantly analysed on a conceptual level [4,18]. For instance, previous studies investigated whether people hold compatibilist views (i.e. the notion that free will can exist in a fully deterministic physical world), or rather hold incompatibilist views (i.e. the notion that free will cannot exist in a fully determined world [19,20]). These, or similar, beliefs are then shown to affect behaviour [21]. While such a conceptual analysis is a highly useful approach to investigate how lay theories relate to specific philosophical stances on free will, it is not informative about laypeople's attitudes towards free will. More broadly, beliefs express agreement to propositional information about free will ('I believe that free will exists.'), while attitudes express a value or liking judgement about free will ('I like free will.'). Although FWB have been shown to affect behaviour [21], it is well established that attitudes are predictive of future behaviour as well [22,23]. Crucially, beliefs and attitudes are not always highly correlated [22], meaning we cannot draw strong conclusions about lay FWAs from the literature on lay FWBs. In this report, we therefore aim to map out attitudes towards free will and related concepts like determinism in the general public. As a first goal, we initially tested whether attitudes towards free will and determinism differ for laypeople (Hypothesis 1, see table 1 for information on all hypotheses and preregistered analyses).

## 1.2. Cultural pressure

In past research on FWB, it has often been assumed that self-reported beliefs accurately represent one's true beliefs. Critically, this relies on the assumption that responses on self-report measures are unbiased. This assumption breaks down if responses are subject to e.g. systematic biases such as socially desirable responding. To date, this remains an untested claim [24], and the issue of biased responding is even more pressing when studying free will attitudes. For instance, past research demonstrated that attitudes are subject to cultural pressure and biased responding [25,26], and socially desirable responding is expected especially if specific responses would violate social norms [25]. In the case of free will, past research has shown that four out of five US citizens believe in the existence of free will [5], suggesting that believing in free will constitutes a social norm. It seems likely that such a norm might exert pressure to report free will as a positive, desirable concept. Thus, a key goal of this research is to establish whether laypeople indeed feel an explicit cultural pressure to like free will, or dislike determinism (Hypotheses 2 and 3), and whether this cultural pressure influences their stated attitudes towards free will (Hypotheses 4–6).

One possible way to address potentially biased responding is to use implicit measures of attitudes (e.g. the implicit association test, IAT, [27,28]), which are less susceptible to socially desirable responding ([28], but see [29]). The IAT uses reaction times to infer implicit attitudes, instead of relying on explicit judgements [28], which has been shown to predict future behaviour [27]. Here, we use both explicit and implicit measures of FWAs. We expect explicit FWAs to be more affected by cultural pressure than implicit FWAs (Hypothesis 4). We further expect low-to-moderate correlations between explicit and implicit FWAs, in line with previous research [30] (Hypothesis 5). However, given the fact that cultural pressure has stronger effects on explicit than on implicit FWAs, this relationship may be influenced by cultural pressure. In other words, we expect cultural pressure to moderate the relationship between explicit and implicit FWAs such that people experiencing strong cultural pressure to like free will, will also have more strongly diverging explicit and implicit FWAs than people who feel no such pressure (Hypothesis 6).

**Table 1.** Hypotheses. We report the sample size in the confirmatory sample ($n$) for each analysis, as well as the effect sizes detectable at 80% power using the proposed statistical test (given $\alpha = 0.05$) at the given sample size. FWA = free will attitude. H1: power analyses were performed separately for implicit and explicit attitude measures. H2/H3: power analyses were performed separately for the two items used to assess cultural pressure (e.g. fw_pressure_pos, and fw_upset_neg). H4/H6: given the larger number of planned tests, we report a range of sample sizes and detectable effect sizes. Power analyses were conducted using R (v. 3.5.1) and GPower (v. 3.1) software.

| hypothesis | sampling plan | | statistical test | interpretation | |
| --- | --- | --- | --- | --- | --- |
| | sample size | power analysis | | positive outcome | negative outcome |
| H1: People differ in their attitude towards free will and determinism. | $n_{\text{explicit}} = 315$, $n_{\text{implicit}} = 1100$ | $d_{\text{explicit}} = 0.16$, $d_{\text{implicit}} = 0.08$ | One-sample, two-sided $t$-test against 0 (explicit) or 0.50 (implicit). | Attitudes are more positive towards free will than towards determinism, or vice versa. | (a) Attitudes towards free will are equal to determinism, or (b) their difference is smaller than the detectable effect size. |
| H2: People feel cultural pressure to value free will. | $n_{\text{fw\_upset\_neg}} = 243$, $n_{\text{fw\_pressure\_pos}} = 268$ | $d_{\text{fw\_upset\_neg}} = 0.18$, $d_{\text{fw\_pressure\_pos}} = 0.17$ | One-sample, two-sided $t$-test versus the midpoint of the scale. | People either feel, or do not feel cultural pressure, depending on the direction of the effect. | (a) People have no strong opinion on cultural pressure items, or (b) cultural pressure is smaller than the detectable effect size. |
| H3: People feel cultural pressure to devalue determinism. | $n_{\text{det\_upset\_pos}} = 243$, $n_{\text{det\_pressure\_neg}} = 267$ | $d_{\text{det\_upset\_pos}} = 0.18$, $d_{\text{det\_pressure\_neg}} = 0.17$ | One-sample, two-sided $t$-test versus the midpoint of the response scale. | People either feel, or do not feel cultural pressure, depending on the direction of the effect. | (a) People have no strong opinion on cultural pressure items, or (b) cultural pressure is smaller than the detectable effect size. |
| H4: Cultural pressure influences explicit FWAs more than implicit FWAs | $75 \leq n \leq 89$ for explicit and $243 \leq n \leq 268$ for implicit FWAs | minimal difference between correlation coefficients between 0.30 and 0.33 | $z$-test between two correlation coefficients. | Influence of cultural pressure on explicit and implicit attitude differs. | (a) There is no difference of cultural pressure effects on explicit and implicit measures, or (b) the difference is smaller than the detectable effect size. |

(Continued.)

**Table 1.** (*Continued.*)

| hypothesis | sampling plan | | statistical test | interpretation | |
|---|---|---|---|---|---|
| | sample size | power analysis | | positive outcome | negative outcome |
| H5: Explicit and implicit attitudes are weakly positively correlated. | $n = 315$ | $r = 0.14$ | Two-sided test of Spearman correlation versus 0. | Explicit and implicit attitudes are either positively or negatively related. | (a) Explicit and implicit attitudes are unrelated, or (b) their relation is weaker than the detectable effect size. |
| H6: Cultural pressure moderates the relationship between explicit and implicit FWAs. | $75 \leq n \leq 89$ | between $f^2 = 0.11$ and $f^2 = 0.13$ | Multiple regression, testing the interaction term (FWA implicit ~ FWA explicit * cultural_pressure) for significance. | Cultural pressure moderates the relation between explicit and implicit attitudes. | (a) Cultural pressure does not modulate relation between explicit and implicit attitudes, or (b) the effect is smaller than the detectable effect size. |

## 1.3. The present investigation

In this registered report, we used a subset of the Attitudes, Identities and Individual Differences (AIID) dataset (https://osf.io/pcjwf/) to investigate attitudes towards free will and related concepts. The AIID dataset is a very large ($N > 200\,000$) planned-missing design dataset in which explicit and implicit attitudes towards 95 different domains (including free will–determinism) were measured, in addition to demographic data and numerous personality questionnaires. The AIID study was designed specifically with reuse potential in mind. Here, we investigated (a) whether people like free will more than determinism, (b) whether there is perceived cultural pressure that might bias responses on FWA items, and (c) assess the effects of such pressure on explicit attitude measures, implicit attitude measures, and on the relationship between them.

# 2. Material and methods

## 2.1. Data and code availability

All data are available upon request from the AIID data curator (Charlie Ebersole, cebersole@virgina.edu), and will be made public in the future (for more information see: https://osf.io/pcjwf/). All code used in the Stage 2 registered report can be accessed here: https://osf.io/jvkhf/.

## 2.2. Exploratory dataset

The AIID dataset is split into two independent datasets. The 'exploratory dataset' is a stratified subset of the full dataset, which uses about 15% of all data. The remaining data are collected in the confirmatory dataset. While the exploratory dataset has been fully accessible to all interested researchers at the Stage 1 of this registered report, the confirmatory dataset has been publicly accessible only in a masked version in which all values were replaced with 1. This allowed us to include *a priori* power analyses for the confirmatory data in the Stage 1 report, without having access to the actual results. Instead, all results reported at Stage 1 were drawn from the exploratory dataset and served only to illustrate the feasibility of the planned analyses. After Stage 1 acceptance, we were granted access to the unmasked confirmatory dataset, and were able to run all planned and exploratory analyses. Hereafter, we report the results in the confirmatory dataset. A detailed description of the analyses and results in the exploratory dataset appears in the Stage 1 registered report proposal document, available at https://osf.io/3kfmw/.

## 2.3. Confirmatory dataset: sample characteristics

The confirmatory AIID dataset contains more than $440\,000$ experimental sessions, acquired between 2005 and 2007. We selected only subjects who completed the free will–determinism evaluation IAT and had no missing IAT data. This resulted in a total sample size of $N = 1100$ for this study (481 female, mean age = 32.24 years, s.d. age = 12.65 years, range = 11–86, country of origin: US 76.27%, CA 4.51%, UK 4.23%, AU 1.27%, NZ 1.27%, other 12.45%). All analyses reported below were performed on this subsample.

Out of the 1100 participants, 122 participants were flagged as having poor performance on the IAT according to the 'exclude_iat' variable in the dataset. Analyses were performed on the full sample of 1100 participants, since this criterion was not included as an exclusion criterion in the preregistration. However, excluding participants with poor performance did not change any of the conclusions.

## 2.4. Procedure

Participants registered online for participation and were then asked to provide detailed demographic information (including sex, age, education, religiosity and political identity). They were assigned a unique user ID and were asked to evaluate 1 out of 95 different domains. These domains included concrete items like 'skirts versus pants', as well as abstract items like 'religion versus atheism', and even scientific theories like 'evolution versus creationism'. The controversy about the latter demonstrates that people can have strong attitudes even towards highly abstract concepts. Here, we focus on the subset of participants assigned to evaluate two other abstract concepts, namely free will

and determinism. All of these participants completed an IAT assessing implicit associations between 'free will–determinism' and 'positive–negative'. In the IAT, these associations are indexed using reaction time measures, and past research has demonstrated that this provides a reliable and valid measure of implicit attitudes [27,31–33] across various domains (e.g. race, sexual orientation). The IATs in the current study were administered and scored following standard IAT procedures [31]. The following words were used to describe free will: 'intention', 'freedom', 'choice', and to describe determinism: 'fixed', 'destined', 'arranged'. Before starting the IAT, each participant was given this list of words. The positive category was described using the words: 'attractive', 'fabulous', 'delightful', 'glorious', 'likable', 'pleasing', while the negative category was described using the words: 'annoy', 'scorn', 'grotesque', 'horrific', 'disaster', 'noxious'.

Besides the IAT, participants were also given between 27 and 29 possible self-report evaluation items about free will and determinism, out of a pool of 79 self-report attitude items. This included items such as valence, liking, gut feelings, actual feelings, confidence and attitude stability (for a full list see https://osf.io/atymr/, and for the exact wording of each item see https://osf.io/3sg5e/). Here, we will focus on assessing self-reported explicit attitudes towards free will and determinism and self-reported culture pressure to value free will or devalue determinism (for a full description of the variables used see below). The order of the IAT and self-reports was randomized across participants.

After the attitude assessment, participants completed 1 out of 15 individual difference questionnaires, measuring Big 5 personality traits, self-monitoring, need for cognition, and other traits. We had no strong *a priori* hypotheses regarding these personality measures for this report. For full details on the dataset used, including the full description of the IAT protocol, please see https://osf.io/atymr/.

## 2.5. Registered analyses

A summary of all hypotheses, the sampling plan and statistical approach is included in table 1. Sample sizes and detectable effect sizes differed for each analysis and are described in table 1 as well. Reported results within each hypothesis were corrected for multiple comparisons using false discovery rate (FDR) corrections when appropriate. All analyses were performed using R (v. 3.5.1).

### 2.5.1. Attitudes towards free will and determinism

To test whether participants had positive/negative attitudes towards free will/determinism (Hypothesis 1, see table 1), we assessed implicit and explicit FWAs. In order to assess implicit FWAs, we used the IAT A score (also referred to as Ruscio's A or the probability of superiority [34]), which is of higher psychometric quality than the commonly reported IAT D score [31,35]. In either case, IAT scores reflect the degree to which a participant is quicker to respond when 'free will' shares a response mapping with positive words and 'determinism' with negative words relative to the opposite mapping. An IAT A score of 0.50 would indicate no difference between these two conditions. For completeness, all tests performed on A scores were also performed on D scores, to help relate our results more easily to past research and serve as a robustness test in the current study. In order to assess the reliability of the IAT measure, we computed internal consistencies using the modal strategy for IAT data (i.e. Spearman–Brown corrected split-half correlations, e.g. [31]). We then assessed the convergent and divergent validity of the IAT by performing a multiple regression (following the approach established by [36]), simultaneously predicting IAT scores from: gut feelings, actual feelings and other people's feelings. These three predictors reflect self-rated intuitive reactions, deliberative reactions, and preferences we are aware of in others. The first two items were rated separately for free will and determinism on a 10-point scale (from 'strongly negative' to 'strongly positive'). To match the IAT scores, we therefore calculated difference scores subtracting the determinism scores from the free will scores (*gut_diff* and *actual_diff*). Preferences in others (*others_prefer*) were rated for free will relative to determinism on a 7-point scale (from 'strongly prefer determinism over free will' to 'strongly prefer free will over determinism'). If the IAT was mostly reflecting fast, intuitive evaluations, we would expect the relation to gut feelings to be stronger than the relations to actual feelings and other people's feelings.

In order to assess explicit FWAs, participants rated both free will and determinism separately on a valence scale ('How positive or negative do you feel towards free will/determinism?', 10-point scale, from 'strongly negative' to 'strongly positive'). To make the explicit measures correspond with the IAT measures, which provide relative measures for the association of free will–determinism with positive–negative, we again calculated a difference score (valence$_{\text{free will}}$ - valence$_{\text{determinism}}$, *val_diff*), with positive values indicating relatively higher valence for free will and negative values indicating

**Table 2.** Descriptive statistics of responses to four cultural pressure items. Percentages of participant agreeing and disagreeing to each item were compared to the midpoint of the scale with two-sided one-sample $t$-tests. $M$ = mean, s.d. = standard deviation, Md = median, IQR = inter-quartile range, $d$ = Cohen's $d$. $p$-Values are corrected for multiple comparisons within each hypothesis using FDR.

| | % agree | % disagree | $t$ (d.f.) | $p$ | $M$ (s.d.) | Md (IQR) | $d$ |
|---|---|---|---|---|---|---|---|
| fw_upset_neg | 30.86 | 69.14 | −7.18 (242) | <0.001 | 2.82 (1.47) | 2 (2) | 0.46 |
| fw_pressure_pos | 79.48 | 20.52 | 11.34 (267) | <0.001 | 4.51 (1.46) | 5 (2) | 0.69 |
| det_upset_pos | 40.33 | 59.67 | −2.93 (242) | 0.004 | 3.22 (1.48) | 3 (2) | 0.19 |
| det_pressure_neg | 65.92 | 34.08 | 4.79 (266) | <0.001 | 3.93 (1.48) | 4 (2) | 0.29 |

relatively higher valence for determinism. While this variable is the closest to the implicit attitudes measured using the IAT, we also repeated all analyses using self-reported preferences for free will versus determinism (*prefer*) as control analyses. Responses to this item ranged from 'Strongly prefer free will over determinism' to 'Strongly prefer determinism over free will'. Although this item is less similar to the IAT than *val_diff*, more subjects responded to this item, making for a larger overall sample size and higher statistical power.

### 2.5.2. Cultural pressure to value free will and/or devalue determinism

In order to test whether subjects felt pressure to value free will positively (Hypothesis 2), we separately assessed responses to two different items: 'No one gets upset if people say bad things about free will.' (*fw_upset_neg*), and 'There is cultural pressure to think positive things about free will.' (*fw_pressure_pos*). Participants responded to both items on a 6-point scale ('strongly disagree', 'disagree', 'slightly disagree', 'slightly agree', 'agree', 'strongly agree'). The benefit of using both items is that one is reverse-coded (*fw_upset_neg*), while the other is not (*fw_pressure_pos*). Furthermore, the response scales can be interpreted meaningfully, in that subjects can be classified into agreeing versus disagreeing. Similarly, in order to test if subjects felt cultural pressure to devalue determinism (Hypothesis 3), we independently assessed responses to two analogous items: 'No one gets upset if people say good things about determinism.' (*det_upset_pos*), and 'There is cultural pressure to think negative things about determinism.' (*det_pressure_neg*). Responses were on the same 6-point scale as before.

For each of the four items, we first computed descriptive statistics (see table 2 for mean, median, standard error of the mean and inter-quartile range) and plotted histograms. Next, we computed the percentage of subjects agreeing to an item, by summing up the number of 'somewhat agree', 'agree' and 'strongly agree' responses, and the percentage of subjects disagreeing with an item, by summing up the number of 'somewhat disagree', 'disagree' and 'strongly disagree' responses. Then, in order to test whether there was significant dis/agreement with any of these items, we performed a two-sided, one-sample $t$-test using the raw response data against the scale's midpoint (which indicates neither agreement not disagreement, i.e. indifference) for each of the four items.

### 2.5.3. The relation between implicit, explicit FWAs and cultural pressure

In a next step, we aimed at assessing whether culture pressure had a stronger influence on explicit compared to implicit attitudes (Hypothesis 4), and whether explicit and implicit attitudes were correlated (Hypothesis 5). To test Hypothesis 4, we computed pairwise Spearman correlations between the four cultural pressure items and both IAT A scores and *val_diff* scores, and subsequently compared the size of these correlations whenever the correlation with either IAT A or *val_diff* was significant. To test Hypothesis 5, we computed Spearman correlations between IAT A and *val_diff* scores.

### 2.5.4. Modulatory effects of cultural pressure on the relation between implicit and explicit FWAs

Next, we assessed to which degree explicit FWAs provide valid information about the supposedly less biased implicit FWAs. That is, does cultural pressure create a divergence between explicit and implicit measures? This should be the case if the former are more affected by cultural pressure than the latter. If this were the case, explicit FWAs should be a better predictor of implicit FWAs for participants that report no cultural pressure than for participants that do report strong cultural pressure (Hypothesis 6). To test this hypothesis, we ran four multiple regression models predicting implicit attitudes (IAT A) from

explicit attitudes (*val_diff*) and cultural pressure items, one for each item (*fw_upset_neg*, *fw_pressure_pos*, *det_upset_pos*, *det_pressure_neg*). Running four separate models was necessary due to the planned missing data design of the AIID dataset. If we found a significant interaction between explicit attitudes and cultural pressure items, this would demonstrate that the relation between explicit and implicit attitude measures is moderated by cultural pressure.

### 2.5.5. Exploratory analyses

The large sample size of this study allowed us to assess the effects of demographic variables on FWAs. There have been previous attempts to assess demographic effects on FWBs, with mixed results [4,5]. Although we had no clear *a priori* hypotheses about the effect of demographic variables, we ran an exploratory analysis that tested for potential relationships between FWAs and age, sex, citizenship, education, race, ethnicity, income, occupation, religion, religiosity and political identity. We estimated a linear mixed model with random effects for citizenship and religion, and with age, sex, education, race, income, religiosity and political identity as predictors. The same approach was used for both explicit and implicit FWAs.

In a similar vein, we correlated explicit and implicit FWAs with the many personality questionnaires employed in this study. These included: Balanced Inventory of Desirable Responding, Bayesian Racism, Belief in a Just World, Big 5 Inventory, Humanitarianism–Egalitarianism, Intuition about Controllability and Awareness of Thoughts, Need for Cognition, Need for Cognitive Closure, Personal Need for Structure, Protestant Ethics, Right-Wing Authoritarianism, Rosenberg Self-Esteem, Self-Monitoring, Social Dominance Orientation, Spheres of Control (for full details see https://osf.io/4dyqz/). There is some prior evidence for a relationship between extraversion and FWBs [36], but this cannot be used to draw strong conclusions about FWAs. We thus had no *a priori* hypotheses about these correlations but aimed at generating novel hypotheses for future research.

Lastly, we were interested in understanding why people might have positive/negative attitudes towards free will in more detail. Responses on the valence scale (e.g. 'strongly positive') accurately describe free will attitudes, yet they provide no indication of *why* people might e.g. have a positive attitude towards free will. In order to explore this issue and generate hypotheses for future research, we performed additional analyses. The AIID dataset contains a number of items that might provide more insights: gut feeling ('strongly negative' to 'strongly positive', 10-point scale), how much free will/determinism are part of one's identity ('How much is free will/determinism part of your self-concept?', 'none at all' to 'very much', 6-point scale), and whether liking or disliking free will/determinism is an important part of one's self-concept ('Being accepting/rejecting of free will/determinism is important to my self-concept.', 'strongly disagree' to 'strongly agree', 6-point scale). For each of these variables, we first computed difference scores (*gut_diff*, *identity_diff* and *self_concept_diff*) by subtracting the ratings for determinism from the ratings for free will. We then computed bivariate Spearman correlations between these items and *val_diff*. In addition to this preregistered analysis, we also ran an additional non-preregistered linear regression predicting valence difference scores from gut difference, identity difference and self-concept difference scores. The rationale behind this non-preregistered analysis was to explore the degree of unique variance in valence difference scores explained by these three variables.

# 3. Results

## 3.1. Measuring implicit and explicit attitudes towards free will

Before assessing potential cultural pressure on FWAs, we first assessed whether participants had positive/ negative attitudes towards free will and determinism (Hypothesis 1). Regarding implicit FWAs, IAT A scores ($M = 0.72$, s.d. $= 0.14$) were significantly larger than the midpoint of 0.50, $t_{1099} = 51.48$, $p < 0.001$, $d = 1.55$, indicating that participants associated free will more strongly with positive words than determinism (figure 1*a*). Split-half reliability of IAT A scores was 0.84, indicating good internal consistency. Regarding convergent and divergent validity, a multiple regression analysis showed that IAT A scores were predicted positively by gut feelings (*gut_diff*), $t_{971} = 2.45$, $p = 0.022$, and by actual feelings (*actual_diff*), $t_{971} = 3.88$, $p < 0.001$, but not by other people's preferences (*others_prefer*), $t_{971} = 1.21$, $p = 0.227$.

Regarding explicit FWAs, *val_diff* scores ($M = 3.40$, s.d. $= 3.44$) were significantly larger than zero, $t_{314} = 17.52$, $p < 0.001$, $d = 0.99$, indicating that participants explicitly valued free will more highly than determinism (figure 1*b*). Alternative measures used as control analyses (IAT D for implicit measures, and self-reported preferences for explicit measures; see Material and methods) correlated highly with

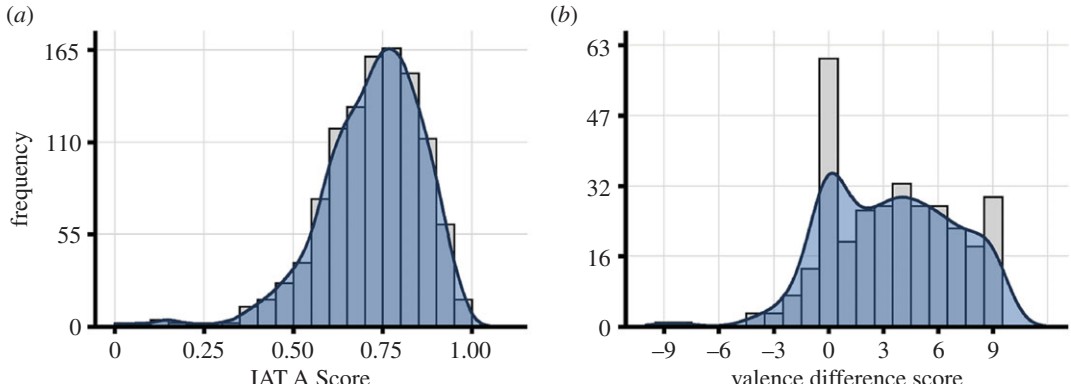

**Figure 1.** Free will attitudes. (*a*) IAT A scores for the free will/determinism. Values above 0.50 indicate stronger associations between free will and positive words than between determinism and positive words. (*b*) Difference valence scores for free will and determinism. Positive values indicate higher valence for free will than for determinism.

their implicit ($r = 0.94$, $p < 0.001$) and explicit counterparts ($r = 0.67$, $p < 0.001$). Overall, using these other dependent variables did not substantially affect the reported results.

### 3.1.1. Non-preregistered analyses

As an additional, non-preregistered analysis, we also computed the percentage of subjects who valued free will and determinism positively (valence scores greater than 5) and negatively (valence scores less than or equal to 5), by looking at both valence scores separately. This revealed that more participants valued free will positively (88.61%) than negatively (11.39%), whereas more people valued determinism negatively (64.04%) than positively (35.96%).

## 3.2. Self-reported cultural pressure

We then assessed whether participants reported an explicit pressure to de/value free will or determinism (Hypotheses 2 and 3) by assessing responses to four different cultural pressure items (see table 2 for descriptive statistics and figure 2 for histograms of responses). In the current dataset, subjects reported significant cultural pressure to value free will positively, and significant cultural pressure to devalue determinism (table 2). However, the pressure on free will was stronger than the pressure on determinism, *fw_upset_neg* versus *det_upset_pos*, $t_{483.99} = -2.98$, $p = 0.003$, $d = 0.27$, *fw_pressure_pos* versus *det_pressure_neg*, $t_{532.94} = 4.58$, $p < 0.001$, $d = 0.40$. Overall, these results suggest that participants experienced cultural pressure to value free will, and that this pressure was stronger for free will than for determinism.

### 3.2.1. Non-preregistered analyses

As an additional non-preregistered analysis, we also performed our planned analyses on the inverse cultural pressure items: i.e. *fw_upset_pos* ('No one gets upset if people say good things about free will.'), *det_upset_neg* ('No one gets upset if people say bad things about determinism.'), *fw_pressure_neg* ('There is cultural pressure to think negative things about free will.'), *det_pressure_pos* ('There is cultural pressure to think positive things about determinism.'). Results in these analyses confirmed our main findings (see https://osf.io/jvkhf/ for more details).

## 3.3. Cultural pressure effects on explicit and implicit attitudes

We then evaluated whether cultural pressure had a stronger influence on explicit compared to implicit attitude measures (Hypothesis 4) by first computing pairwise Spearman correlation coefficients between all four cultural pressure items, IAT A scores, and *val_diff* scores (table 3). This also allowed us to test if explicit and implicit attitudes were correlated (Hypothesis 5).

We found that implicit and explicit attitudes were moderately correlated ($r = 0.25$, $p < 0.001$). However, in contrast with our hypothesis, none of the cultural pressure items were correlated with either *val_diff* scores, all $p \geq 0.240$, or IAT A scores, all $p \geq 0.974$. We did not find any significant differences between correlations with implicit versus explicit attitude measures for any of the cultural pressure items, all $z \leq 1.74$, $p \geq 0.32$.

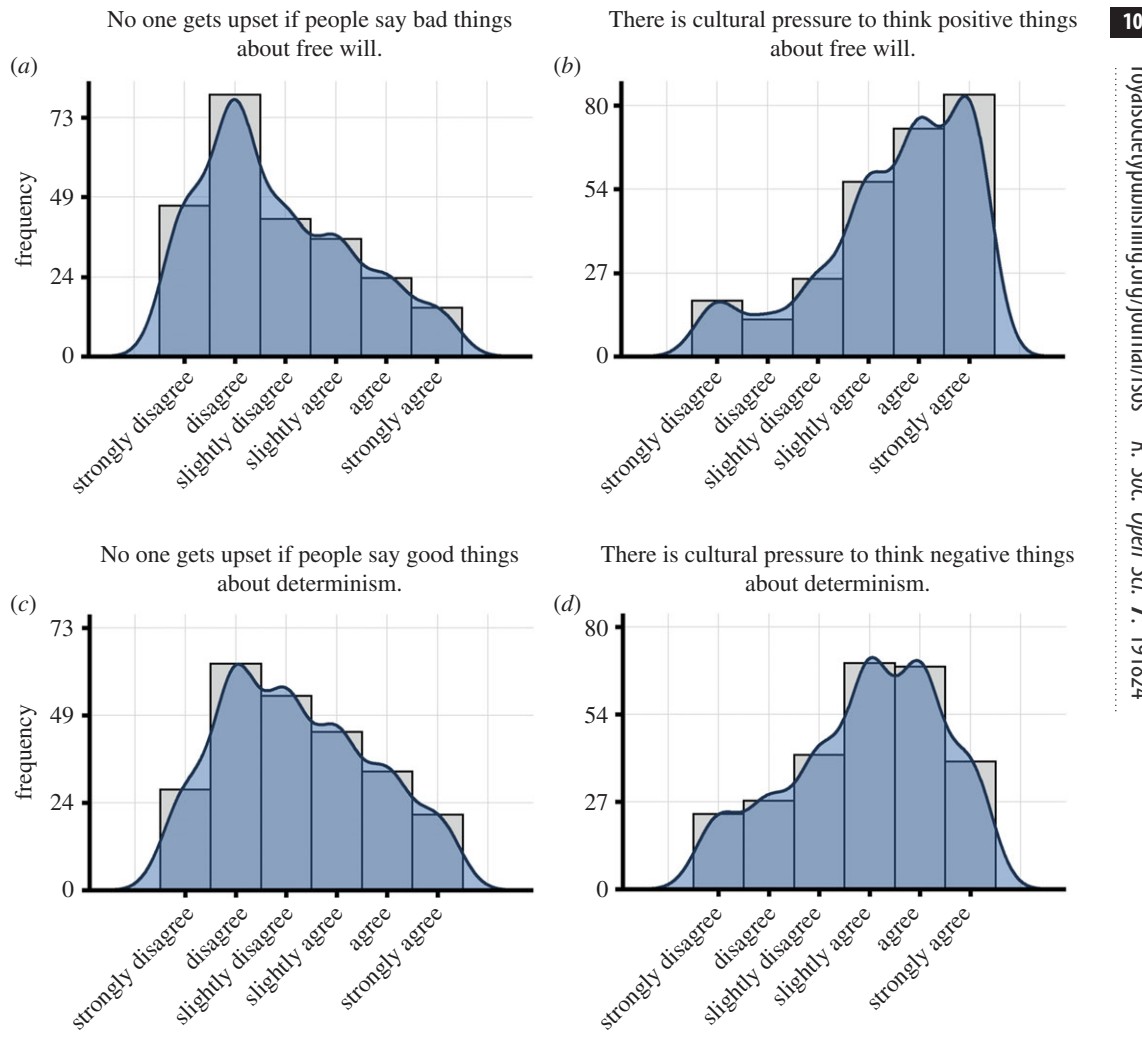

**Figure 2.** Cultural pressure items. Histograms of responses to cultural pressure items towards free will (*a*,*b*) and determinism (*c*,*d*).

**Table 3.** Correlations between implicit (IAT A) and explicit (*val_diff*) attitudes and cultural pressure. Cells below the diagonal show Spearman correlation coefficients. Cells above the diagonal show *p*-values. *p*-Values relevant to our hypotheses were corrected for multiple comparisons using FDR. *p*-Values not related to our hypotheses were not corrected and are printed in italics.

|  | val_diff | IAT A | fw_upset_neg | fw_pressure_pos | det_upset_pos | det_pressure_neg |
|---|---|---|---|---|---|---|
| val_diff | — | <0.001 | 0.992 | 0.974 | 0.240 | 0.974 |
| IAT A | 0.25 | — | 0.974 | 0.974 | 0.992 | 0.992 |
| fw_upset_neg | 0.00 | 0.03 | — | n.a. | *<0.001* | n.a. |
| fw_pressure_pos | 0.09 | −0.03 | n.a. | — | n.a. | *<0.001* |
| det_upset_pos | −0.22 | 0.01 | 0.49 | n.a. | — | n.a. |
| det_pressure_neg | −0.07 | 0.01 | n.a. | 0.55 | n.a. | — |

## 3.4. Cultural pressure and biased responding

Next, we assessed whether cultural pressure moderated the relationship between explicit and implicit attitudes (Hypothesis 6). In contrast with our hypothesis, even though *val_diff* scores positively predicted IAT A scores in all four models, all $p \leq 0.006$, none of the interactions between *val_diff* scores and cultural pressure scores were significant, all $p \geq 0.476$ (table 4). Thus, cultural pressure did not lead to more divergent explicit and implicit attitudes.

**Table 4.** Regression models predicting IAT A from val_diff and cultural pressure. All predictors were mean-centred so that the val_diff effects reflect the effect of val_diff while cultural pressure (*cult_press*) was kept constant at its mean value rather than at 0. *p*-Values were corrected using FDR for testing multiple cultural pressure items, separately for the two tests of interest.

| | $\beta$ | s.e. | $t$ | $p$ |
|---|---|---|---|---|
| **fw_upset_neg** | | | | |
| val_diff | 0.015 | 0.005 | 3.14 | 0.005 |
| val_diff × cult_press | 0.001 | 0.003 | 0.35 | 0.726 |
| **fw_pressure_pos** | | | | |
| val_diff | 0.011 | 0.004 | 2.96 | 0.005 |
| val_diff × cult_press | −0.002 | 0.002 | −1.02 | 0.476 |
| **det_upset_pos** | | | | |
| val_diff | 0.014 | 0.005 | 2.98 | 0.005 |
| val_diff × cult_press | 0.003 | 0.003 | 0.93 | 0.476 |
| **det_pressure_neg** | | | | |
| val_diff | 0.011 | 0.004 | 2.85 | 0.006 |
| val_diff × cult_press | −0.003 | 0.002 | −1.15 | 0.476 |

## 3.5. Exploratory analyses

### 3.5.1. Effects of demographic variables

Of all the demographic variables assessed, we only found a positive relation between age and FWA, both for implicit measures, $F_{1,595} = 37.00$, $p < 0.001$, and explicit measures, $F_{1,162} = 16.10$, $p < 0.001$, suggesting that older participants had more positive attitudes towards free will than younger participants. No other demographic variables were predictive of FWAs.

### 3.5.2. Effects of personality traits

After correcting for multiple comparisons (FDR correction), no significant correlations were found between personality traits and explicit and implicit FWAs (table 5), all *p*-values between 0.32 and 0.99.

### 3.5.3. The relation between free will attitudes and gut feelings, self-concept and identity

In order to explore why people might have specific attitudes towards free will and determinism, we computed bivariate correlations between gut feelings towards free will/determinism (*gut_diff*), the degree to which free will/determinism is part of one's identity (*identity_diff*), the degree to which dis/liking free will/determinism is an important part of one's self-concept (*self_concept_diff*), and explicit FWAs (*val_diff*) (table 6). We found *val_diff* scores to correlate positively with *gut_diff* ($r = 0.65$, $p < 0.001$), suggesting that attitudes towards free will might be partly driven by intuitive evaluations. However, *val_diff* scores were also correlated with *identity_diff* and *self_concept_diff* scores (all $p < 0.001$).

#### 3.5.3.1. Non-preregistered analyses

To identify the unique contribution of each variable to valence judgements, we ran a non-preregistered regression model predicting *val_diff* scores from *gut_diff*, *identity_diff* and *self_concept_diff* scores. This indicated that *val_diff* were predicted significantly by *identity_diff* scores, $t_{116} = 8.18$, $p < 0.001$, and to a lesser extent *self_concept_diff* scores, $t_{116} = 2.33$, $p = 0.032$, but not by gut feeling scores, $t_{116} = 1.82$, $p = 0.071$. This demonstrates that FWAs might be more closely linked to identity and self-concept, than they are to gut feelings about free will.

# 4. Discussion

In this registered report, we investigated attitudes towards free will and determinism in a large lay sample for the first time. Using data from the AIID project, we analysed both explicit and implicit

**Table 5.** Correlations between attitudes and personality traits. Spearman correlations between explicit (*val_diff*) and implicit (IAT A) free will attitudes and personality traits measured in the AIID dataset. BFI: Big Five Inventory with subscales Extraversion (E), Conscientiousness (C), Neuroticism (N), Agreeableness (A) and Openness (O). NFCC: Need for Cognitive Closure with subscales Predictability (P), Decisiveness (D), Close-Mindedness (C), Order (O) and Ambiguity (A). BIDR: Balanced Inventory of Desirable Responding with subscales Impression Management (IM) and Self-Deception (SDE). BJW: Belief in Just World. BRS: Bayesian Racism. HE: Humanitarian-Egalitarianism. ICAT: Intuitions about Controllability and Awareness of Thoughts with subscales Others (O) and Self (S). NFC: Need for Cognition. PE: Protestant Ethic. PNS: Personal Need for Structure. RSE: Rosenberg Self-Esteem. RWA: Right-Wing Authoritarianism. SDO: Social Dominance Orientation. SM: Self-Monitoring. SOC: Spheres of Control with subscales Interpersonal Control (IC) and Personal Efficacy (PE). For full details see https://osf.io/4dyqz/. Uncorrected *p*-values are reported. None of the correlations remained significant after correcting for multiple comparisons using FDR.

| | val_diff | | | IAT A | | |
|---|---|---|---|---|---|---|
| | *r* | *n* | *p* | *r* | *n* | *p* |
| BFI_E | −0.08 | 16 | 0.763 | 0.08 | 51 | 0.578 |
| BFI_C | −0.15 | 16 | 0.571 | −0.08 | 50 | 0.601 |
| BFI_N | 0.26 | 16 | 0.330 | −0.05 | 51 | 0.718 |
| BFI_A | 0.11 | 23 | 0.606 | 0.07 | 55 | 0.626 |
| BFI_O | −0.19 | 25 | 0.354 | 0.29 | 57 | 0.030 |
| NFCC_P | −0.53 | 15 | 0.041 | −0.19 | 42 | 0.238 |
| NFCC_D | 0.45 | 15 | 0.094 | 0.19 | 41 | 0.224 |
| NFCC_C | −0.25 | 15 | 0.367 | −0.35 | 41 | 0.023 |
| NFCC_O | 0.05 | 14 | 0.860 | 0.21 | 57 | 0.119 |
| NFCC_A | n.a. | 0 | n.a. | n.a. | 0 | n.a. |
| BIDR_IM | 0.09 | 13 | 0.765 | 0.06 | 43 | 0.693 |
| BIDR_SDE | 0.63 | 13 | 0.020 | −0.02 | 55 | 0.858 |
| BJW | −0.52 | 17 | 0.031 | −0.06 | 55 | 0.644 |
| BRS | 0.18 | 10 | 0.610 | 0.13 | 42 | 0.412 |
| HE | 0.34 | 14 | 0.230 | 0.23 | 50 | 0.111 |
| ICAT_O | −0.27 | 11 | 0.422 | −0.06 | 52 | 0.687 |
| ICAT_S | 0.22 | 10 | 0.538 | 0.01 | 41 | 0.928 |
| NFC | 0.64 | 13 | 0.020 | 0.11 | 47 | 0.460 |
| PE | −0.49 | 13 | 0.086 | −0.12 | 42 | 0.448 |
| PNS | −0.16 | 13 | 0.605 | 0.04 | 52 | 0.790 |
| RSE | −0.17 | 18 | 0.493 | −0.06 | 59 | 0.626 |
| RWA | −0.18 | 19 | 0.466 | −0.18 | 64 | 0.150 |
| SDO | −0.30 | 18 | 0.234 | 0.00 | 60 | 0.986 |
| SM | 0.13 | 18 | 0.606 | 0.16 | 50 | 0.270 |
| SOC_IC | 0.32 | 15 | 0.245 | 0.09 | 40 | 0.579 |
| SOC_PE | 0.43 | 15 | 0.106 | −0.08 | 45 | 0.623 |

attitudes towards free will and determinism. Attitudes towards free will were generally positive, while attitudes towards determinism were generally negative. This finding was robust to changes in how explicit and implicit attitudes were computed from the data. We further showed that people experience cultural pressure to value free will, and to devalue determinism. Contrary to our hypotheses, this pressure did not seem to affect free will attitudes.

## 4.1. Mapping free will attitudes in the general public

There is much prior research demonstrating that free will beliefs are dynamic, i.e. that they can be manipulated experimentally [7,16,21,37]. Such experimental manipulations often lead only to small

**Table 6.** Correlations between explicit attitudes (*val_diff*), gut feelings (*gut_diff*), identity (*identity_diff*) and self-concept (*self_concept_diff*). Cells below the diagonal show Spearman correlation coefficients. Cells above the diagonal show *p*-values. *p*-Values relevant to our hypotheses were corrected for multiple comparisons using FDR. *p*-Values not related to our hypotheses were not corrected and are printed in italics.

|  | val_diff | gut_diff | identity_diff | self_concept_diff |
|---|---|---|---|---|
| val_diff | — | <0.001 | <0.001 | <0.001 |
| gut_diff | 0.65 | — | *<0.001* | *<0.001* |
| identity_diff | 0.78 | 0.60 | — | *<0.001* |
| self_concept_diff | 0.57 | 0.46 | 0.55 | — |

effects, and there is little research directly addressing why changing free will beliefs is so difficult. Furthermore, free will belief manipulations probably affect different people to a varying degree, and we currently do not know which variables make a person susceptible to such manipulations. One factor that might partly explain these open questions is people's attitudes towards free will. A person with highly positive attitudes towards free will is likely to be less affected by FWB manipulations than a person with more negative attitudes towards free will. In order to test this hypothesis, we first need to map out and understand FWAs in the general public, which has not been done to date, however.

Our results show that both implicit and explicit attitudes towards free will are strongly positive in the general public. Only 11.39% of the participants explicitly valued free will negatively, while 34% chose the most positive available attitude rating on the scale, which mirrors similar findings for FWBs [4,5]. Furthermore, a lower proportion of participants explicitly valued determinism positively (35.96%), which again mirrors weaker belief in determinism in previous studies [5]. Interestingly, the mode of the explicit attitude distribution was nevertheless 0, indicating that many participants indicated to value free will and determinism to an equal degree. The same was not true for the implicit attitude distribution, which had a clearly positive mode instead. Thus, although many people indicated that they valued free will and determinism to an equal degree, implicit measures suggested that they in fact valued free will more strongly.

Additionally, our data demonstrate that there is considerable variance in FWAs (both explicit and implicit). Taken together, these results suggest that FWAs are a potential mediator for FWB manipulations, and might help explain why such manipulations are generally weak. Of course, these conclusions are tentative at the moment and will have to be confirmed empirically. Specifically, further research will need to directly assess the relation of FWBs and FWAs, and whether FWAs act as a mediator for FWB manipulations.

## 4.2. Cultural pressure and free will attitudes

One untested assumption in the free will belief literature is that responses to free will questionnaires are unbiased by cultural pressure. Here, we showed that in the domain of free will attitudes, participants indeed report cultural pressure to value free will, as well as a cultural pressure to devalue determinism. We further show that cultural pressure is stronger on free will than it is on determinism. Given this fact, one might expect that this cultural pressure affects attitudes in some measurable way. To investigate this possibility, we tested if more self-reported pressure to value free will led to more positive FWAs and whether more self-reported pressure led to more diverging explicit and implicit FWAs, since we expected cultural pressure to affect explicit attitudes more strongly than implicit ones. Both of these analyses yielded no significant results, however, leading us to conclude that cultural pressure did not affect FWA measures in our sample.

Although unexpected, there are some potential explanations for this finding. One option is that the cultural pressure measure did not fully capture the desired construct (biased/pressured responding). In order to assess cultural pressure, we only used two items that were rather blunt (e.g. 'There is cultural pressure to think positive things about free will.'). Biased responding is clearly a wider concept than just self-rated cultural pressure, and is often measured using whole scales like the Balanced Inventory of Desirable Responding (BIDR) [38], a 40-item scale that measures both 'impression management' and 'self-deceptive enhancement'. Such measures are more nuanced and of higher psychometric quality than the two cultural pressure items used in our main analysis.

While the BIDR was also part of the AIID dataset, we chose to use the cultural pressure items because the available sample size ($214 < n < 233$) was much higher than for the BIDR ($13 < n < 55$), and because these items were directly related to free will and determinism, while the BIDR is a much more general trait measure. However, the BIDR impression management and self-deceptive enhancement scales were included in the exploratory analysis correlating FWAs with personality traits, and we assessed BIDR scores in additional non-preregistered analyses. Impression management, which is arguably closer to the cultural pressure items used in the main analysis, did not correlate with either explicit ($r = 0.09$, $p = 0.77$, $n = 13$) or implicit ($r = 0.06$, $p = 0.69$, $n = 43$) attitudes. Thus, even this alternative measure of 'cultural pressure' showed no significant results. In contrast, self-deceptive enhancement did correlate with explicit attitudes ($r = 0.63$, $p = 0.02$, $n = 13$), but not with implicit attitudes ($r = −0.02$, $p = 0.86$, $n = 55$). Thus, it might be that people who tend to hold exaggerated positive self-descriptions report more positive feelings towards free will, relative to determinism. This exploratory finding is based on a very small sample size though and should be interpreted with caution until is it replicated in an independent, larger sample. Clearly, more research on this is needed, but this already demonstrates that free will attitudes might still be biased, just not by self-rated cultural pressure.

## 4.3. Exploratory analyses

In additional registered exploratory analyses, we found that age was positively related to free will attitudes. Interestingly, this finding is not consistent with a recent report that free will beliefs do not change with age [5]. This points to an interesting dissociation between free will beliefs and free will attitudes with increasing age that should be addressed in the future.

## 4.4. Directions for future research

Despite the many interesting findings in the current dataset, there remain several key open questions that should be addressed. First, the methods for measuring implicit attitudes towards free will need to be further refined. While the IAT is a much used and well validated instrument, its design only allows the assessment of two concepts in direct opposition. The IAT treats free will and determinism as two endpoints on a single attitude scale: the more you value free will, the less you value determinism, and vice versa. We know from free will belief research, however, that believing in free will does not automatically translate into disbelieving determinism [5], in fact one can believe in both at the same time (compatibilism, [1]). Thus, it might be that people also value both free will and determinism positively, yet the IAT is ill-suited to detect this pattern of free will related attitudes. Additionally, we found IAT scores to be more strongly related to deliberate thoughts on free will, than they are to gut feelings on free will, a pattern of results that we did not expect. Lastly, the words used to describe free will (intention, freedom, choice), and determinism (fixed, destined, arranged) have many uses that have little to do with the philosophical concepts of either free will or determinism, and might not precisely capture them. For instance, 'arranged' is often used to describe arranging a meeting, which has little to do with physical determinism. Taken together, this suggests that future research should invest effort into developing alternative measures of implicit attitudes towards free will.

Second, methods for measuring explicit attitudes towards free will and determinism should also be refined, and the precise wording of the attitude items might need to be revised. Additional effort should be invested into the precise wording of the attitude items. The AIID project used a wide range of different explicit attitude items, and here we focused on valence items ('How positive or negative do you feel towards free will/determinism?'). For research specifically focusing on free will, it might be that other items (e.g. 'Having free will is important to me') are better suited to capture free will attitudes. Additionally, in order to make explicit and implicit attitude measures comparable here, we computed a difference score from free will and determinism valence ratings (*val_diff*). While this is useful from a methodological perspective, this procedure suffers from some of the same issues as the free will IAT. Just as the IAT cannot detect people who value both free will and determinism positively, as some compatibilists might do, the *val_diff* items cannot do so either. In the future, explicit attitude items for free will and determinism should be investigated separately, to more easily identify compatibilist participants.

Third, future work should focus on investigating free will beliefs and attitudes simultaneously, in order to describe and explain their interactions. Here, we focused on attitudes only, but in order to understand belief attitude interaction, e.g. whether free will attitudes mediate effects of free will belief manipulations on behaviour, we need to assess beliefs and attitudes within the same participants.

This would also allow us to test whether free will attitudes are conditional on specific free will beliefs (e.g. 'I can only value free will if I believe in it'), or vice versa. Relatedly, data reported here have been acquired between 2005 and 2007, while much of the work on free will beliefs has been performed about a decade later [4,5]. Attitudes can change across time, which further emphasizes the necessity to acquire beliefs and attitudes simultaneously if we are to understand how these two aspects of lay views on free will interact.

## 4.5. Conclusion

In this study, we show that attitudes towards free will are highly positive in the general public, mirroring previous findings showing a strong belief in free will. Although we show that there is considerable cultural pressure to value free will, we were unable to show any effects of that pressure on free will attitudes.

We further provide evidence on the distribution of free will attitudes for the first time. In the past, research on free will exclusively focused on free will beliefs, largely ignoring the equally important attitudes. Our study thus opens the door to a whole new line of research on the determinants and effects of free will attitudes. Our results are also interesting for the free will belief literature. An untested assumption in the FWB literature was that FWB measures are unbiased by e.g. socially desirable responding. Our results on free will attitudes are not easily transferable to the FWB literature, but we are taking them as a cautiously optimistic finding for this literature. Showing that attitudes are not strongly affected by cultural pressure at least makes it less likely that such effects will be present for free will beliefs. Surely, this conclusion is tentative, and more research specifically designed to assess this issue is necessary.

Ethics. This Stage 2 Registered Report uses data from the previously published Attitudes, Identities and Individual Differences (AIID) dataset (https://osf.io/pcjwf/). The original study and data acquisition was approved by the IRB of the University of Virginia. No new data were acquired for the present investigation.

Data accessibility. The raw data used in the Stage 1 registered report can be accessed here: https://osf.io/pcjwf/. The approved Stage 1 protocol can be accessed here: https://osf.io/3kfmw/. All data are available upon request from the AIID data curator (Charlie Ebersole, cebersole@virginia.edu) and will be made public in the future (for more information see: https://osf.io/pcjwf/). All code used in the Stage 2 registered report can be accessed here: https://osf.io/jvkhf/.

Authors' contributions. E.C. developed the study concept, analysis pipeline, performed analyses on the exploratory and confirmatory dataset, and revised the draft. C.G.-G. developed the study concept, analysis pipeline, performed analyses on the exploratory dataset, wrote the article draft, and revised the draft. I.H. developed the analysis pipeline, and revised the draft. S.B. developed the study concept, analysis pipeline, and revised the draft. D.W. developed the study concept, analysis pipeline, performed analyses on the exploratory dataset, wrote the article draft, and revised the draft.

Competing interests. The authors declare no competing interests.

Funding. E.C. was supported by a Research Foundations Flanders (FWO) grant (FWO18/PDO/049). C.G.-G. was supported by the Special Research Fund of Ghent University (BOF.GOA.2017.0002.03), and the European Union's Horizon 2020 research and innovation programme under the Marie Sklodowska-Curie grant agreement no. 835767. D.W. was supported by the FWO (FWO.KAN.2019.0023.01), and the European Union's Horizon 2020 research and innovation programme under the Marie-Skłodowska-Curie grant agreement no. 665501.

Acknowledgements. We would like to thank the rest of the AIID team (Sean Hughes, Calvin K. Lai, Charles R. Ebersole, Jordan Axt, Brian A. Nosek) for making their data publicly available and pushing the boundaries of open research practices.

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
