## [Reviewer comments · Royal Society Open Science]

Review History

Decision letter (RSOS-191723.R0)

07-Oct-2019

Dear Dr Wisniewski,

I write you in regards to manuscript RSOS-191723 entitled "Cultural pressure and biased responding in free will attitudes" which you submitted to Royal Society Open Science.

We routinely triage submissions for scientific soundness, clarity and general adherence to the Registered Reports guidelines. For submissions that have promise but are not yet suitable for in-depth Stage 1 review, we offer feedback to help authors maximise the chances that reviewers will respond positively to a resubmission.

We have concluded that your submission is not yet suitable for in-depth review and has therefore been rejected at this time, but we believe it will be suitable once several issues are addressed. We therefore invite a resubmission. Further comments from the Associate Editor may be found at the end of this letter.

If you wish to revise your manuscript in light of the below comments please submit your

manuscript as a new submission and mention this previous manuscript ID in your covering letter. You should also provide a detailed response to the below comments in the cover letter.

Thank you for considering Royal Society Open Science for the publication of your registered report.

Kind regards,
Anita Kristiansen
Editorial Coordinator
Royal Society Open Science
openscience@royalsociety.org

on behalf of Professor Chris Chambers (Registered Reports Editor, Royal Society Open Science)
openscience@royalsociety.org

Associate Editor Comments to Author:

Associate Editor

Comments to the Author:

The following point needs to be addressed before the manuscript can be sent to in-depth review:

1. The protocol needs to include clearly specified and enumerated hypotheses and a specific sampling plan (e.g. power analysis or Bayesian alternative, either analytically or produced using simulations) associated with every statistical test (or tests) that will interrogate every hypothesis. In addition, a clearer overall mapping is required between the hypotheses, sampling plan(s), statistical tests and prospective interpretations of the different possible outcomes. We recommend including the hypotheses in a list either at the end of the Introduction or later in the manuscript, and then make clear the specific test or test component that will test each hypothesis in the subsequent analysis plan and sampling plan. This could also be presented as a table that indicates the hypothesis, sampling plan (e.g. power analysis), precise analysis or analyses, and prospective interpretation of different outcomes.

Author's Response to Decision Letter for (RSOS-191723.R0)

See Appendix A.

RSOS-191824.R0

Review form: Reviewer 1

Do you have any ethical concerns with this paper?

No

Recommendation?

Accept in principle

Comments to the Author(s)

Stage 1 Registered Report RSOS-191824

"Cultural pressure and biased responding in free will attitudes"

We would be grateful if you can assess the following aspects of the submission:

1. The scientific validity of the research question(s): This research has the potential to make a valuable contribution to the literature and the experimental design is well-suited to allow the team to shed light on the issue that interests them.
2. The logic, rationale, and plausibility of the proposed hypotheses: Both the logic and rationale behind the proposal are sound and the proposed hypotheses are plausible.
3. The soundness and feasibility of the methodology and analysis pipeline (including statistical power analysis where applicable): The methodology and analysis pipeline are both sound and feasible.
4. Whether the clarity and degree of methodological detail would be sufficient to replicate the proposed experimental procedures and analysis pipeline: Give the proposed design – namely, using a publicly available data set to explore attitudes about free will and determinism – it will be easy for subsequent teams to attempt to replicate the findings.
5. Whether the authors provide a sufficiently clear and detailed description of the methods to prevent undisclosed flexibility in the experimental procedures or analysis pipeline: The team’s description is very thorough and clear. This should prevent undisclosed flexibility in the experimental procedures or analysis pipeline.
6. Whether the authors have considered sufficient outcome-neutral conditions (e.g. absence of floor or ceiling effects; positive controls; other quality checks) for ensuring that the results obtained are able to test the stated hypotheses: The team’s experimental design is well-suited for testing their stated hypotheses.

Overview:

These researchers plan to extend the work that has been done in free will beliefs by looking instead (or additionally) at free will attitudes. Using a large, publicly available data set, the team wants to explore people’s explicit and implicit attitudes about free will and determinism. Given that little, if any, work has been done on this issue, the present research has the potential to make an important contribution to the literature. That said, I nevertheless had some thoughts I wanted to share with the editors/team.

Random Thoughts:

First, it would have been helpful had the team provided all of the precise wordings used in the original study for free will and determinism. These are technical terms of philosophical art, so the devil is always in the terminological details. Take, for instance, the following two examples cited by the team: “strongly prefer free will over determinism” and “strongly prefer determinism over free will.” Here the key terms go undefined – which is problematic given that research suggests that people have competing understandings of both free will and determinism. While some people construe them as opposites (as they are construed here) others view them as compatible. This is why researchers have tried to carefully define these key terms in a way that allows us to know how participants were using the term.

Second, while I appreciate the team’s interest in attitudes (vs. beliefs) about free will and determinism, the notion strikes me as somewhat odd. Free will and determinism are not like money or art or sports – where I can understand why people value these things differentially. But, consider, for instance, determinism. Here the attitudinal question is strange. What would it mean to value determinism or indeterminism? You might as well ask people whether they value string theory or the multiverse. These are all theories about the nature of the universe – it seems odd to treat them as targets of valuing. Just to be clear: I am not suggesting that this project isn’t interesting and shouldn’t be undertaken. I am merely pointing out that one might have a priori reasons for wondering whether it makes sense to explore people’s attitudes about free will and determinism. This is something the team would need to address in the introduction.

Third, the preliminary findings are hard to interpret. Take, for instance, the fact that participants seem to like free will more than determinism and feel more pressure to like free will more than determinism. It's unclear whether this has anything to do with free will per se. Instead, it could merely be driven by the fact that most people think we have free will and most people think determinism is false. I take it that it is a general truth that there is pressure to believe things that most people take to be true and less pressure to believe things most people take to be false. If that's right, there is nothing unique going on when it comes to free will and determinism. For any widely held and cherished belief x , there is pressure both to believe and to like x .

Review form: Reviewer 2 (Jonathan Schooler)

Do you have any ethical concerns with this paper?

No

Recommendation?

Accept with minor revision

Comments to the Author(s)

See attached (Appendix B).

Decision letter (RSOS-191824.R0)

20-Dec-2019

Dear Dr Wisniewski

On behalf of the Editors, I am pleased to inform you that your Stage 1 Registered Report RSOS-191824 entitled "Cultural pressure and biased responding in free will attitudes" has been accepted in principle for publication in Royal Society Open Science subject to minor revision in accordance with the referee and editor suggestions. Please find their comments at the end of this email.

The reviewers and handling editors have recommended publication, but also suggest some minor revisions to your manuscript. Therefore, I invite you to respond to the comments and revise your manuscript.

Please you submit the revised version of your manuscript within 30 days (i.e. by the 20th January 2020). If you do not think you will be able to meet this date please let me know immediately.

Full author guidelines can be found here <https://royalsocietypublishing.org/rsos/registered-reports#ReviewerGuideRegRep>.

Kind regards,
Anita Kristiansen
Editorial Coordinator
Royal Society Open Science
openscience@royalsociety.org

on behalf of Professor Chris Chambers (Subject Editor, Royal Society Open Science)
openscience@royalsociety.org

Associate Editor Comments to Author (Professor Chris Chambers):

Associate Editor: 1

Comments to the Author:

Two expert reviewers have now appraised the Stage 1 manuscript. Both assessments are overall reasonably positive, but nevertheless note a number of areas that will require careful attention, including clarification (and justification) of the study rationale and implications of the potential findings, and validity of the outcome measures. Addressing the latter point may require the inclusion of additional preregistered analyses.

Reviewer comments to Author:

Reviewer: 1

Comments to the Author(s)

Stage 1 Registered Report RSOS-191824

"Cultural pressure and biased responding in free will attitudes"

We would be grateful if you can assess the following aspects of the submission:

1. The scientific validity of the research question(s): This research has the potential to make a valuable contribution to the literature and the experimental design is well-suited to allow the team to shed light on the issue that interests them.
2. The logic, rationale, and plausibility of the proposed hypotheses: Both the logic and rationale behind the proposal are sound and the proposed hypotheses are plausible.
3. The soundness and feasibility of the methodology and analysis pipeline (including statistical power analysis where applicable): The methodology and analysis pipeline are both sound and feasible.
4. Whether the clarity and degree of methodological detail would be sufficient to replicate the proposed experimental procedures and analysis pipeline: Give the proposed design – namely, using a publicly available data set to explore attitudes about free will and determinism – it will be easy for subsequent teams to attempt to replicate the findings.
5. Whether the authors provide a sufficiently clear and detailed description of the methods to prevent undisclosed flexibility in the experimental procedures or analysis pipeline: The team's description is very thorough and clear. This should prevent undisclosed flexibility in the experimental procedures or analysis pipeline.
6. Whether the authors have considered sufficient outcome-neutral conditions (e.g. absence of floor or ceiling effects; positive controls; other quality checks) for ensuring that the results obtained are able to test the stated hypotheses: The team's experimental design is well-suited for testing their stated hypotheses.

Overview:

These researchers plan to extend the work that has been done in free will beliefs by looking instead (or additionally) at free will attitudes. Using a large, publicly available data set, the team wants to explore people's explicit and implicit attitudes about free will and determinism. Given that little, if any, work has been done on this issue, the present research has the potential to make an important contribution to the literature. That said, I nevertheless had some thoughts I wanted to share with the editors/team.

Random Thoughts:

First, it would have been helpful had the team provided all of the precise wordings used in the original study for free will and determinism. These are technical terms of philosophical art, so the devil is always in the terminological details. Take, for instance, the following two examples cited by the team: "strongly prefer free will over determinism" and "strongly prefer determinism over free will." Here the key terms go undefined – which is problematic given that research suggests that people have competing understandings of both free will and determinism. While some people construe them as opposites (as they are construed here) others view them as compatible. This is why researchers have tried to carefully define these key terms in a way that allows us to know how participants were using the term.

Second, while I appreciate the team's interest in attitudes (vs. beliefs) about free will and determinism, the notion strikes me as somewhat odd. Free will and determinism are not like money or art or sports – where I can understand why people value these things differentially. But, consider, for instance, determinism. Here the attitudinal question is strange. What would it mean to value determinism or indeterminism? You might as well ask people whether they value string theory or the multiverse. These are all theories about the nature of the universe – it seems odd to treat them as targets of valuing. Just to be clear: I am not suggesting that this project isn't interesting and shouldn't be undertaken. I am merely pointing out that one might have a priori reasons for wondering whether it makes sense to explore people's attitudes about free will and determinism. This is something the team would need to address in the introduction.

Third, the preliminary findings are hard to interpret. Take, for instance, the fact that participants seem to like free will more than determinism and feel more pressure to like free will more than determinism. It's unclear whether this has anything to do with free will per se. Instead, it could merely be driven by the fact that most people think we have free will and most people think determinism is false. I take it that it is a general truth that there is pressure to believe things that most people take to be true and less pressure to believe things most people take to be false. If that's right, there is nothing unique going on when it comes to free will and determinism. For any widely held and cherished belief x , there is pressure both to believe and to like x .

Reviewer: 2

Comments to the Author(s)
See attached

Author's Response to Decision Letter for (RSOS-191824.R0)

See Appendix C.

Decision letter (RSOS-191824.R1)

21-Feb-2020

Dear Dr Wisniewski

On behalf of the Editor, I am pleased to inform you that your Manuscript RSOS-191824.R1 entitled "Cultural pressure and biased responding in free will attitudes" has been accepted in principle for publication in Royal Society Open Science.

You may now progress to Stage 2 and complete the study as approved. Before commencing data collection we ask that you:

- 1) Update the journal office as to the anticipated completion date of your study.
- 2) Register your approved protocol on the Open Science Framework (<https://osf.io/>) or other recognised repository, either publicly or privately under embargo until submission of the Stage 2 manuscript. Please note that a time-stamped, independent registration of the protocol is mandatory under journal policy, and manuscripts that do not conform to this requirement cannot be considered at Stage 2. The protocol should be registered unchanged from its current approved state, with the time-stamp preceding implementation of the approved study design. We recommend using the dedicated portal for registration of Stage 1 RRs at <https://osf.io/rr>

Following completion of your study, we invite you to resubmit your paper for peer review as a Stage 2 Registered Report. Please note that your manuscript can still be rejected for publication at Stage 2 if the Editors consider any of the following conditions to be met:

- The results were unable to test the authors' proposed hypotheses by failing to meet the approved outcome-neutral criteria.
- The authors altered the Introduction, rationale, or hypotheses, as approved in the Stage 1 submission.
- The authors failed to adhere closely to the registered experimental procedures. Please note that any deviations from the approved experimental procedures must be communicated to the editor immediately for approval, and prior to the completion of data collection. Failure to do so can result in revocation of in-principle acceptance and rejection at Stage 2 (see complete guidelines for further information).
- Any post-hoc (unregistered) analyses were either unjustified, insufficiently caveated, or overly dominant in shaping the authors' conclusions.
- The authors' conclusions were not justified given the data obtained.

We encourage you to read the complete guidelines for authors concerning Stage 2 submissions at <https://royalsocietypublishing.org/rsos/registered-reports#ReviewerGuideRegRep>. Please especially note the requirements for data sharing, reporting the URL of the independently registered protocol, and that withdrawing your manuscript will result in publication of a Withdrawn Registration.

Please note that Royal Society Open Science will introduce article processing charges for all new submissions received from 1 January 2018. Registered Reports submitted and accepted after this date will ONLY be subject to a charge if they subsequently progress to and are accepted as Stage 2 Registered Reports. If your manuscript is submitted and accepted for publication after 1 January 2018 (i.e. as a full Stage 2 Registered Report), you will be asked to pay the article processing charge, unless you request a waiver and this is approved by Royal Society Publishing. You can find out more about the charges at <https://royalsocietypublishing.org/rsos/charges>. Should you have any queries, please contact openscience@royalsociety.org.

Once again, thank you for submitting your manuscript to Royal Society Open Science and we look forward to receiving your Stage 2 submission. If you have any questions at all, please do not hesitate to get in touch. We look forward to hearing from you shortly with the anticipated submission date for your stage two manuscript.

on behalf of Professor Chris Chambers (Registered Reports Editor, Royal Society Open Science)
openscience@royalsociety.org

Author's Response to Decision Letter for (RSOS-191824.R1)

See Appendix D.

RSOS-191824.R2 (Revision)

Review form: Reviewer 1

Is the manuscript scientifically sound in its present form?

Yes

Are the interpretations and conclusions justified by the results?

Yes

Is the language acceptable?

Yes

Do you have any ethical concerns with this paper?

No

Have you any concerns about statistical analyses in this paper?

No

Recommendation?

Accept with minor revision

Comments to the Author(s)

Referee Report:

Cultural pressure and biased responding in free will attitudes

RSOS-191824.R2

Preliminary Questions:

1. The data are able to test the authors' proposed hypotheses by passing the approved outcome-neutral criteria (such as absence of floor and ceiling effects or success of positive controls or other quality checks).
2. The introduction, rationale and stated hypotheses are the same as the approved Stage 1 submission.
3. The authors adhered precisely to the registered experimental procedures.
4. The unregistered exploratory statistical analyses used by the authors are justified, methodologically sound, and informative.
5. Setting aside some of the concerns I raised below, the authors' conclusions are justified given the data.

General Assessment:

Free will beliefs have recently received a lot of attention in the literature from experimental philosophy, social psychology, and cognitive science. These authors make a novel contribution to this area of research by switching focus from people's beliefs about free will to their attitudes about free will. As such, this paper represents an important step in a new direction that broadens the scope of research on the role that free will plays in people's lives. It also opens up a fertile new area for empirical investigation. I nevertheless have several concerns about how attitudes about free will have been operationalized, which in turn problematizes the conclusions drawn from the findings.

Despite these concerns, I think the paper merits publication given the novelty of the authors' focus (and besides, the fact that they had a very large sample and didn't find any correlations between free will attitudes and a battery of personality measures is a very important finding that in itself makes an important contribution to the field). The concerns I raise below are not necessarily concerns I think the authors need to address for the purposes of publishing this paper. They are concerns that I think would need to be addressed in future work now that they have opened up this new line of investigation.

As will be clear, several of the issues I raise are based on the extant data and variables the authors used for their analyses. For a preliminary investigation, it makes perfect sense that they would use some of the variables from the IAT literature (even if they are suboptimal). Despite the problems with some of these variables, I think the authors are to be applauded for pushing research on the psychology of free will in a new and important direction. But moving forward, more care would need to be taken to improve upon the measures that are used here to explore people's attitudes about free will, determinism, and related constructs.

Issues and Suggestions:

First, it sounds very odd and unnatural to say, "I like free will," or "I don't like free will." This isn't how one would naturally express the fact that one values free will. The "I like..." locution makes more sense when talking about fruits, vegetables, music genres, people, etc. It sounds odd when it comes to free will. Something like "I value free will" or "Having free will is important to me" both sound much more natural. In my own case, I don't think we have free will. But I wouldn't say I don't like it (even if I don't necessarily value it). Given that this entire project is about people's attitudes about free will, I don't think this is a marginal issue. The items used for measuring these attitudes are crucial. And I worry that some of the items are worded in odd and unnatural ways.

Second, there is a related concern when it comes to the relationship between free will beliefs and free will attitudes. If I don't believe that free will exists, then it's hard to know what sense it makes to say that "I like free will." It would be akin to not believing in souls or ghosts, but saying that I nevertheless like souls or ghosts. If I don't believe that free will exists, what sense does it make for me to either like it or dislike it? Instead, it seems like the issue is whether or not I would prefer to have free will regardless of whether I actually have it. But to get at this, one would need

a different approach than the one adopted by the authors. A nested or branching approach would work better. Something like: First, you ask whether they believe in free will. If yes, you ask how much they value it? If no, you ask how much it matters to [or bothers] them that they don't have it. Or something along those lines...

Third, the IAT items the authors used are problematic. Free will is operationalized using "intention", "freedom", and "choice", while determinism is operationalized using "fixed", "destined", "arranged". While it's true that the exercise of free will often involves intentions, freedom, and choice, these concepts are dissociable from free will in myriad ways. For instance, someone like me who doesn't believe in free will might nevertheless believe in intentions, choices, and other kinds of freedom (e.g., political). So, it's not clear to me that these items shed light on free will per se. As for the determinism items, while fixed and destined seem good enough proxies, arranged is positively bizarre (since there are tons of uses of the term which have nothing to do with determinism). In short, I am not sure how illuminating the results of the IAT analysis are given that the terms used are problematic. I realize that the authors did not design these items, but insofar as their analysis is built on them, they inherit whatever problems are associated with them.

Fourth, the authors claim:

"Besides the IAT, participants were also given between 27 and 29 possible self-report evaluation items about free will and determinism, out of a pool of 79 self-report attitude items. This included items such as valence, liking, gut feelings, actual feelings, confidence, and attitude stability. Here, we will focus on assessing self-reported explicit attitudes towards free will and determinism and self-reported culture pressure to like/dislike free will or determinism (for a full description of the variables used see below)."

Perhaps I missed it, but I couldn't find the list of these variables. They should be included somewhere in the paper.

Fifth, the authors claim:

"In order to assess explicit FWAs, participants rated both free will and determinism separately on a valence scale (10-point scale, from "strongly negative" to "strongly positive")."

But they do not provide the precise wording that the participants received for these items. What were the directions? Where they asked, "What are your attitudes or feelings towards the following terms?"

Sixth, I am worried that the authors have made it impossible for compatibilists to accurately report their explicit attitudes. Consider, for instance, the following:

"Responses to this item ranged from "Strongly prefer free will over determinism" to "Strongly prefer determinism over free will".

Compatibilists don't see any conflict between free will and determinism so it's not clear how they are supposed to respond to this kind of forced ranking of preferences. Indeed, some compatibilists think that without determinism, we can't be free. So, for people who have these views – and surely there are some people with these views – it's not clear these answer choices capture their actual preferences.

Decision letter (RSOS-191824.R2)

Dear Dr Wisniewski:

On behalf of the Editor, I am pleased to inform you that your Stage 2 Registered Report RSOS-191824.R2 entitled "Cultural pressure and biased responding in free will attitudes" has been deemed suitable for publication in Royal Society Open Science subject to minor revision in accordance with the referee suggestions. Please find the referees' comments at the end of this email.

The reviewers and Subject Editor have recommended publication, but also suggest some minor revisions to your manuscript. Therefore, I invite you to respond to the comments and revise your manuscript.

Please also ensure that all the below editorial sections are included where appropriate -- if any section is not applicable to your manuscript, please can we ask you to nevertheless include the heading, but explicitly state that the heading is inapplicable. An example of these sections is attached with this email.

- Ethics statement

- Data accessibility

[http://datadryad.org/submit?journalID=RSOS&manu=\(Document not available\)](http://datadryad.org/submit?journalID=RSOS&manu=(Document not available))

- Competing interests

- Authors' contributions

- Acknowledgements

- Funding statement

Because the schedule for publication is very tight, it is a condition of publication that you submit the revised version of your manuscript within 7 days (i.e. by the 30-Jul-2020). If you do not think you will be able to meet this date please let me know immediately.

Please note that Royal Society Open Science will introduce article processing charges for all new submissions received from 1 January 2018. Registered Reports submitted and accepted after this date will ONLY be subject to a charge if they subsequently progress to and are accepted as Stage 2 Registered Reports. If your manuscript is submitted and accepted for publication after 1 January 2018 (i.e. as a full Stage 2 Registered Report), you will be asked to pay the article processing charge, unless you request a waiver and this is approved by Royal Society Publishing. You can find out more about the charges at <https://royalsocietypublishing.org/rsos/charges>. Should you have any queries, please contact openscience@royalsociety.org.

on behalf of Professor Chris Chambers
 (Registered Reports Editor, Royal Society Open Science)
openscience@royalsociety.org

Associate Editor Comments to Author (Professor Chris Chambers):

Associate Editor: 1

Comments to the Author:

The Stage 2 manuscript was returned to one of the original Stage 1 reviewers -- the other reviewer was unavailable. The reviewer's assessment is positive and mirrors my own (non-specialist) opinion having also read the manuscript again. The reviewer offers a thoughtful critique of the conceptual basis of the research, in large part as a reflection on future steps. I think this is a useful contribution, especially given that the journal's open review policy. However, in revising, please confine any responses to the issues raised (and consideration of limitations) to the Discussion, as it is not possible to alter or relitigate the rationale and methods approved at Stage 1. Provided the authors are able to respond thoroughly to all points raised (including provision of some additional detail -- see the reviewer's point 5), either through revision or rebuttal, a revised Stage 2 manuscript is likely to be accepted without requiring further in-depth review.

Comments to Author:

Reviewer: 1

Comments to the Author(s)

Referee Report:

Cultural pressure and biased responding in free will attitudes

RSOS-191824.R2

Preliminary Questions:

1. The data are able to test the authors' proposed hypotheses by passing the approved outcome-neutral criteria (such as absence of floor and ceiling effects or success of positive controls or other quality checks).
2. The introduction, rationale and stated hypotheses are the same as the approved Stage 1 submission.
3. The authors adhered precisely to the registered experimental procedures.
4. The unregistered exploratory statistical analyses used by the authors are justified, methodologically sound, and informative.

5. Setting aside some of the concerns I raised below, the authors' conclusions are justified given the data.

General Assessment:

Free will beliefs have recently received a lot of attention in the literature from experimental philosophy, social psychology, and cognitive science. These authors make a novel contribution to this area of research by switching focus from people's beliefs about free will to their attitudes about free will. As such, this paper represents an important step in a new direction that broadens the scope of research on the role that free will plays in people's lives. It also opens up a fertile new area for empirical investigation. I nevertheless have several concerns about how attitudes about free will have been operationalized, which in turn problematizes the conclusions drawn from the findings.

Despite these concerns, I think the paper merits publication given the novelty of the authors' focus (and besides, the fact that they had a very large sample and didn't find any correlations between free will attitudes and a battery of personality measures is a very important finding that in itself makes an important contribution to the field). The concerns I raise below are not necessarily concerns I think the authors need to address for the purposes of publishing this paper. They are concerns that I think would need to be addressed in future work now that they have opened up this new line of investigation.

As will be clear, several of the issues I raise are based on the extant data and variables the authors used for their analyses. For a preliminary investigation, it makes perfect sense that they would use some of the variables from the IAT literature (even if they are suboptimal). Despite the problems with some of these variables, I think the authors are to be applauded for pushing research on the psychology of free will in a new and important direction. But moving forward, more care would need to be taken to improve upon the measures that are used here to explore people's attitudes about free will, determinism, and related constructs.

Issues and Suggestions:

First, it sounds very odd and unnatural to say, "I like free will," or "I don't like free will." This isn't how one would naturally express the fact that one values free will. The "I like..." locution makes more sense when talking about fruits, vegetables, music genres, people, etc. It sounds odd when it comes to free will. Something like "I value free will" or "Having free will is important to me" both sound much more natural. In my own case, I don't think we have free will. But I wouldn't say I don't like it (even if I don't necessarily value it). Given that this entire project is about people's attitudes about free will, I don't think this is a marginal issue. The items used for measuring these attitudes are crucial. And I worry that some of the items are worded in odd and unnatural ways.

Second, there is a related concern when it comes to the relationship between free will beliefs and free will attitudes. If I don't believe that free will exists, then it's hard to know what sense it makes to say that "I like free will." It would be akin to not believing in souls or ghosts, but saying that I nevertheless like souls or ghosts. If I don't believe that free will exists, what sense does it make for me to either like it or dislike it? Instead, it seems like the issue is whether or not I would prefer to have free will regardless of whether I actually have it. But to get at this, one would need a different approach than the one adopted by the authors. A nested or branching approach would work better. Something like: First, you ask whether they believe in free will. If yes, you ask how much they value it? If no, you ask how much it matters to [or bothers] them that they don't have it. Or something along those lines...

Third, the IAT items the authors used are problematic. Free will is operationalized using "intention", "freedom", and "choice", while determinism is operationalized using "fixed", "destined", "arranged". While it's true that the exercise of free will often involves intentions,

freedom, and choice, these concepts are dissociable from free will in myriad ways. For instance, someone like me who doesn't believe in free will might nevertheless believe in intentions, choices, and other kinds of freedom (e.g., political). So, it's not clear to me that these items shed light on free will per se. As for the determinism items, while fixed and destined seem good enough proxies, arranged is positively bizarre (since there are tons of uses of the term which have nothing to do with determinism). In short, I am not sure how illuminating the results of the IAT analysis are given that the terms used are problematic. I realize that the authors did not design these items, but insofar as their analysis is built on them, they inherit whatever problems are associated with them.

Fourth, the authors claim:

"Besides the IAT, participants were also given between 27 and 29 possible self-report evaluation items about free will and determinism, out of a pool of 79 self-report attitude items. This included items such as valence, liking, gut feelings, actual feelings, confidence, and attitude stability. Here, we will focus on assessing self-reported explicit attitudes towards free will and determinism and self-reported culture pressure to like/dislike free will or determinism (for a full description of the variables used see below)."

Perhaps I missed it, but I couldn't find the list of these variables. They should be included somewhere in the paper.

Fifth, the authors claim:

"In order to assess explicit FWAs, participants rated both free will and determinism separately on a valence scale (10-point scale, from "strongly negative" to "strongly positive")."

But they do not provide the precise wording that the participants received for these items. What were the directions? Where they asked, "What are your attitudes or feelings towards the following terms?"

Sixth, I am worried that the authors have made it impossible for compatibilists to accurately report their explicit attitudes. Consider, for instance, the following:

"Responses to this item ranged from "Strongly prefer free will over determinism" to "Strongly prefer determinism over free will".

Compatibilists don't see any conflict between free will and determinism so it's not clear how they are supposed to respond to this kind of forced ranking of preferences. Indeed, some compatibilists think that without determinism, we can't be free. So, for people who have these views – and surely there are some people with these views – it's not clear these answer choices capture their actual preferences.

Author's Response to Decision Letter for (RSOS-191824.R2)

See Appendix E.

Decision letter (RSOS-191824.R3)

Dear Dr Wisniewski:

It is a pleasure to accept your manuscript entitled "Cultural pressure and biased responding in free will attitudes" in its current form for publication in Royal Society Open Science.

on behalf of Professor Chris Chambers (Subject Editor)
openscience@royalsociety.org

Appendix A

Comments from the Editor:

The protocol needs to include clearly specified and enumerated hypotheses and a specific sampling plan (e.g. power analysis or Bayesian alternative, either analytically or produced using simulations) associated with every statistical test (or tests) that will interrogate every hypothesis. In addition, a clearer overall mapping is required between the hypotheses, sampling plan(s), statistical tests and prospective interpretations of the different possible outcomes. We recommend including the hypotheses in a list either at the end of the Introduction or later in the manuscript, and then make clear the specific test or test component that will test each hypothesis in the subsequent analysis plan and sampling plan. This could also be presented as a table that indicates the hypothesis, sampling plan (e.g. power analysis), precise analysis or analyses, and prospective interpretation of different outcomes.

We fully agree that the mapping of planned analyses onto our hypotheses was not clear enough. We now include a table (Table 1) summarizing all hypotheses, sampling plans (including power analyses), planned analyses, and interpretation of positive/negative outcomes. All hypotheses are now numbered, and referred to in the text as such (e.g. Hypothesis 2, Table 1). We believe this now provides the necessary clarity to the reader.

Appendix B

Review MS# RSOS-191824

Summary- This proposed study will use a pre-existing data set to examine the relationship between individuals' beliefs in free will and their attitudes about free will. It will compare both explicit attitudes and implicit attitudes about free will and investigate the degree to which they are associated with socially desirable responding. The six primary hypotheses that will be explored are: 1) People differ in their attitudes towards free will and determinism 2) people feel cultural pressure to like free will, 3) people feel cultural pressure to dislike determinism, 4) cultural pressure influences explicit free will attitudes more than implicit ones, 5) implicit and explicit attitudes towards free will are weakly correlated and 6) Cultural pressure moderates the relationship between explicit and implicit free will attitudes. Implicit and explicit attitudes towards free will will also be related to a host of other individual difference measures such as the big 5 and need for cognition.

Evaluation- This study takes advantage of a remarkable data set in order to gain new insights into an aspect of free will (people's attitudes towards it) that has largely been overlooked. The conjecture that attitudes about free will may differ substantially from belief in free will seems highly plausible as does the speculation that people will feel cultural pressure to like free will and dislike determinism. Furthermore, the planned analyses seem appropriate and I am enthusiastic about the fact that the entire project is being pre-registered before the results are fully tallied.

Although I am generally enthusiastic about this project there are a few aspects of it that I am uncertain about. First although I agree in principle that attitudes towards free will are important to distinguish from belief in free will I do not think the authors do a sufficient job of characterizing what liking or disliking the concept of free will would mean. Would disliking free will imply that a person felt burdened by the responsibilities of choice? Would liking free will mean that people are pleased that they have control? The authors do very little to help the reader understand the implications of the construct that they are seeking to explicate.

I am also a bit concerned about what differences between implicit and explicit measures of attitudes towards free will might mean. The primary prediction is that explicit measures will be more affected by cultural pressures than implicit measures. However, perhaps people don't have a well defined implicit attitude towards free will. This seems plausible to me as just as researchers have not dedicated much research towards investigating attitudes towards free will it seems that people may not have devoted much thought about the value of the concept and thus may not have a particularly robust implicit attitude. Accordingly discrepancies between implicit and explicit measures might just reflect undeveloped implicit attitudes. Explicit attitudes may exist but might be simply developed on the fly. "Now that I think about it I guess free will is an pretty important idea" if so differences between implicit and explicit attitudes might simply reflect the lack of a stable implicit measure. Such a concern would be addressed if implicit measures correlated well with some measures and explicit more with others. Alternatively if implicit measures of free will correlated well internally but were well distinguished from other implicit attitudes this would at least reassure me that it is a stable

construct. Anything the authors can do to shore up the validity of the implicit attitude measure of free will and determinism would be helpful if they come to demonstrate that implicit and explicit attitudes reliably differ.

Jonathan Schooler

Appendix C

Reviewer #1

Overview:

These researchers plan to extend the work that has been done in free will beliefs by looking instead (or additionally) at free will attitudes. Using a large, publicly available data set, the team wants to explore people’s explicit and implicit attitudes about free will and determinism. Given that little, if any, work has been done on this issue, the present research has the potential to make an important contribution to the literature. That said, I nevertheless had some thoughts I wanted to share with the editors/team.

We thank the reviewer for the overall positive evaluation of our work, and provide point-by-point responses below.

Random Thoughts:

First, it would have been helpful had the team provided all of the precise wordings used in the original study for free will and determinism. These are technical terms of philosophical art, so the devil is always in the terminological details. Take, for instance, the following two examples cited by the team: “strongly prefer free will over determinism” and “strongly prefer determinism over free will.” Here the key terms go undefined—which is problematic given that research suggests that people have competing understandings of both free will and determinism. While some people construe them as opposites (as they are construed here) others view them as compatible. This is why researchers have tried to carefully define these key terms in a way that allows us to know how participants were using the term.

The reviewer rightly points out that the manuscript is currently missing information on how the concepts of free will and determinism were introduced to participants. During the IAT, the concepts are generally not introduced in much detail. This is true for domains like race, liberty, or evolution, and was also true for free will here. At the beginning of the experiment, participants are told they will have to categorize words belonging to two categories: free will, and determinism. They are then given the list of words used to describe both categories: “Intention, Freedom, Choice” (free will), and “Fixed, Destined, Arranged” (determinism). This gives participants a frame of reference, in which to interpret free will and determinism as concepts, and is a standard procedure for administering IATs.

Importantly, this does not constrain the interpretation of these concepts too much, as the goal here is to look at lay intuitions and attitudes. E.g. giving a precise definition of compatibilism might bias lay views in that direction, which would confound our results. We investigated lay beliefs in the past (Wisniewski, Deutschländer, Haynes, PLoS ONE, 2019), and found them to be somewhat inconsistent. Most people seem to agree to both compatibilist and incompatibilist notions of free will, and in light of these findings, keeping the description of free will neutral with respect to the compatibilism issue seems the best way to ensure unbiased results that best reflect lay beliefs.

We now amended the methods section to describe in more detail how free will and determinism were framed for the participants.

Methods (pages 6/7):

Here, we focus on the subset of participants assigned to evaluate “free will - determinism”. These participants completed an IAT that assessed implicit associations between “free will - determinism” and “positive - negative”.

The following words were used to describe free will: “intention”, “freedom”, “choice”, and to describe determinism: “fixed”, “destined”, “arranged”. Before starting the IAT, each participant

was given this list of words associated with free will and determinism, respectively. The positive category was described using the words: “attractive”, “fabulous”, “delightful”, “glorious”, “likable”, “pleasing”, while the negative category was described using the words: “annoy”, “scorn”, “grotesque”, “horrific”, “disaster”, “noxious”.

Second, while I appreciate the team’s interest in attitudes (vs. beliefs) about free will and determinism, the notion strikes me as somewhat odd. Free will and determinism are not like money or art or sports—where I can understand why people value these things differentially. But, consider, for instance, determinism. Here the attitudinal question is strange. What would it mean to value determinism or indeterminism? You might as well ask people whether they value string theory or the multiverse. These are all theories about the nature of the universe—it seems odd to treat them as targets of valuing. Just to be clear: I am not suggesting that this project isn’t interesting and shouldn’t be undertaken. I am merely pointing out that one might have a priori reasons for wondering whether it makes sense to explore people’s attitudes about free will and determinism. This is something the team would need to address in the introduction.

We thank the reviewer for raising this important point, yet are unsure whether the reviewer believes that having attitudes towards abstract concepts or towards scientific theories is implausible. In either case, we would argue that people can hold attitudes about everything that they have beliefs about, including abstract concepts and theories. Regarding abstract concepts, the AIID dataset itself demonstrates that some of the strongest attitudes (among the 95 domains tested) are towards abstract concepts like forgiveness or order (<https://mmdata.shinyapps.io/AIIDexplorer/>). Regarding scientific theories, the debate of evolutionary theory and creationism, which is quite heated in some regions of the world, demonstrates that people can have strong attitudes towards scientific theories. Interestingly, in the AIID dataset, attitudes towards free will were even more polarized than towards evolutionary theory (<https://mmdata.shinyapps.io/AIIDexplorer/>). We now point out more explicitly that people can have attitudes even towards abstract, scientific concepts in the methods section.

Methods (page 6).

[participants] ... were assigned a unique user ID and were then assigned to evaluate 1 out of 95 different domains. These domains included concrete items like “skirts vs pants”, as well as abstract items like “religion vs atheism”, and even scientific theories like “evolution vs creationism”. The debate about the latter demonstrates that people can even have strong attitudes towards highly abstract scientific theories, which led us to think that the same might be true for “free will vs determinism” as well.

Thus, we believe that people have attitudes towards free will and determinism. As to why we investigate such attitudes, we believe that attitudes might play a key mediating role when it comes to changes in free will beliefs (as stated on page 3). Previous research suggests that free will beliefs can be manipulated experimentally, and might even change in the general public. Some influential scientists claim that free will is “an illusion”, because all our behavior is physically determined. Such statements might reduce belief in free will in the general public. We argue that any such changes critically depend in attitudes towards free will (page 3). If we have a positive attitude towards free will, we will be less likely to revise our beliefs, as compared to if we have a negative attitude. But in order to test this, we need to understand free will attitudes first.

Third, the preliminary findings are hard to interpret. Take, for instance, the fact that participants seem to like free will more than determinism and feel more pressure to like free will more than determinism. It’s unclear whether this has anything to do with free will per se. Instead, it could merely be driven by

the fact that most people think we have free will and most people think determinism is false. I take it that it is a general truth that there is pressure to believe things that most people take to be true and less pressure to believe things most people take to be false. If that's right, there is nothing unique going on when it comes to free will and determinism. For any widely held and cherished belief x, there is pressure both to believe and to like x.

Although we fully agree that widely held beliefs likely exert pressure, it is not our intention to claim any unique process underlying free will attitudes. Importantly, however, the absence of such pressure has been an (implicit) assumption in much of the past free will research: the idea that free will beliefs can be measured accurately using questionnaires critically relies on the assumption that responses to the questions are unbiased. This assumption breaks down if there is cultural pressure to respond in a certain way and by measuring cultural pressure to like/dislike free will, we thus empirically test this assumption for the first time. Should the preliminary findings be confirmed in the full dataset, this would demonstrate the presence of cultural pressure, which would be a key novel finding, necessitating a re-interpretation of past results. That is, the finding that people strongly believe in free will could then be partly attributable to common cultural pressure, possibly demonstrating that the true belief in free will is lower than the one measured. This point is now stated more explicitly in the introduction section.

Introduction (pages 4/5)

In past research on FWB, it has often been assumed that self-reported beliefs accurately represent one's true beliefs. Critically, this relies on the assumption that responses on self-report measures are unbiased. This assumption breaks down if responses are subject to e.g. systematic biases such as socially desirable responding.

... past research has shown that 4 out of 5 US citizens believe in the existence of free will, suggesting that believing in free will constitutes a social norm. It seems likely that such a norm might exert pressure to report free will as a positive, desirable concept.

Reviewer #2

Summary- This proposed study will use a pre-existing data set to examine the relationship between individuals' beliefs in free will and their attitudes about free will. It will compare both explicit attitudes and implicit attitudes about free will and investigate the degree to which they are associated with socially desirable responding. The six primary hypotheses that will be explored are: 1) People differ in their attitudes towards free will and determinism 2) people feel cultural pressure to like free will, 3) people feel cultural pressure to dislike determinism, 4) cultural pressure influences explicit free will attitudes more than implicit ones, 5) implicit and explicit attitudes towards free will are weakly correlated and 6) Cultural pressure moderates the relationship between explicit and implicit free will attitudes. Implicit and explicit attitudes towards free will will also be related to a host of other individual difference measures such as the big 5 and need for cognition.

Evaluation- This study takes advantage of a remarkable data set in order to gain new insights into an aspect of free will (people's attitudes towards it) that has largely been over looked. The conjecture that attitudes about free will may differ substantially from belief in free will seems highly plausible as does the speculation that people will feel cultural pressure to like free will and dislike determinism. Furthermore, the planned analyses seem appropriate and I am enthusiastic about the fact that the entire project is being pre-registered before the results are fully tallied.

We would like to thank you for the overall positive evaluation of our work, and provide point-by-point responses below.

Although I am generally enthusiastic about this project there are a few aspects of it that I am uncertain about. First although I agree in principle that attitudes towards free will are important to distinguish from belief in free will I do not think the authors do a sufficient job of characterizing what liking or disliking the concept of free will would mean. Would disliking free will imply that a person felt burdened by the responsibilities of choice? Would liking free will mean that people are pleased that they have control? The authors do very little to help the reader understand the implications of the construct that they are seeking to explicate.

The reviewer notes that we currently do not characterize in much detail why participants would have a specific attitude towards free will. While participants can indicate whether they like the concept or not, they are indeed not required to give specific reasons for these judgements. There are many reasons for having positive or negative attitudes towards free will (e.g. based on a specific philosophical stance vs. based on an unspecific gut feeling), and the current study was not optimized to fully describe these reasons in detail. We agree that this question is important, and we will raise this issue in the discussion section to clearly point out avenues for future research.

Moreover, based on the reviewer's point, we now also decided to include additional exploratory analyses to further explore this issue with the data at hand. Specifically, the AIID dataset provides a number of different explicit attitude measures that can be used to describe attitudes in more detail. In the main analysis, we restricted ourselves to valence and preference judgments, in order to avoid issues with multiple comparison corrections. But we now also report responses to the following items: gut feelings, the degree to which free will/determinism is part of one's self-concept, and whether positive evaluations are important for one's self-concept. Responses to these items will be correlated with the valence judgements reported in the main analysis. Each of these items could help reveal why a specific attitude is being held, and thus provide a tentative explanation that could be followed up in future research.

Exploratory analyses (pages 18/19)

2.7.3 Exploring why people have positive/negative attitudes towards free will and determinism

In this study, we investigate whether participants have a positive or negative attitudes towards free will, and determinism. For this purpose, we analyze responses on a valence scale, which allows us to determine where people fall on the spectrum between "strongly positive" and "strongly negative". Although this accurately describes free will attitudes, responses on this scale provide no indication of why people might e.g. have a positive attitude towards free will. In order to explore this issue and generate hypotheses for future research, we will perform additional analyses.

The AIID dataset contains a number of items that might provide more insights into why people might have specific attitudes: gut feeling ("strongly negative" to "strongly positive, 10-point scale), how much free will / determinism are part of one's self-concept ("How much is free will / determinism part of your self-concept?", "none at all" to "very much", 6-point scale), and whether liking or disliking free will / determinism is an important part of one's self-concept ("Being accepting/rejecting of Free Will/Determinism is important to my self-concept.", "strongly disagree" to "strongly agree", 6-point scale). We will compute bivariate correlations between these items, and the valence judgements from the main analysis. Due to the fact that no participant responded to all items in this study, the sample size and statistical power will again differ for each bivariate correlation tested here. We will be able to detect small to medium effects

ranging from $r = 0.14$ ($n = 315$) to $r = 0.28$ ($n = 76$). If for instance, we found valence to correlate positively with gut feelings, this might indicate that attitudes are at least partly driven by intuitive evaluations. Clearly, future research optimized to dissociate different reasons for having specific FWAs will be necessary, and these exploratory analyses might help generate hypotheses for such research.

I am also a bit concerned about what differences between implicit and explicit measures of attitudes towards free will might mean. The primary prediction is that explicit measures will be more affected by cultural pressures than implicit measures. However, perhaps people don't have a well defined implicit attitude towards free will. This seems plausible to me as just as researchers have not dedicated much research towards investigating attitudes towards free will it seems that people may not have devoted much thought about the value of the concept and thus may not have a particularly robust implicit attitude. Accordingly discrepancies between implicit and explicit measures might just reflect undeveloped implicit attitudes. Explicit attitudes may exist but might be simply developed on the fly. "Now that I think about it I guess free will is an pretty important idea" if so differences between implicit and explicit attitudes might simply reflect the lack of a stable implicit measure. Such a concern would be addressed if implicit measures correlated well with some measures and explicit more with others.

Alternatively if implicit measures of free will correlated well internally but were well distinguished from other implicit attitudes this would at least reassure me that it is a stable construct. Anything the authors can do to shore up the validity of the implicit attitude measure of free will and determinism would be helpful if they come to demonstrate that implicit and explicit attitudes reliable differ.

We agree that the absence of implicit attitudes could pose a problem to the interpretation of our results, as described above. However, there is evidence that makes this unlikely. The exploratory dataset already demonstrates that people have surprisingly strong attitudes towards free will (page 12, and <https://mmdata.shinyapps.io/AIIDexplorer/>). In the AIID dataset, attitudes were measured towards 95 different domains, and these can be ranked with respect to the strength of implicit attitudes people have. Of all 95 domains, free will – determinism ranks 11th, showing surprisingly strong implicit attitudes. Free will ranks higher than religion – atheism, or conservatives – liberals. Thus, we believe it to be unlikely that people will have no implicit attitudes towards free will.

The second point raised relates to the measure we use to quantify implicit attitudes (IAT). The reviewer is concerned about the reliability and validity of the free will IAT. The IAT has been proven to be a reliable measure of implicit attitudes across many domains (e.g. Greenwald et al. JPSP 2003), and we expect the free will IAT to be a reliable measure too. In order to provide additional evidence for the reliability of the specific free will IAT used here, we will report internal consistencies (Spearman-Brown corrected split half correlations). Thus, even in the unlikely case that we find no implicit attitudes towards free will, we will know whether this stems from an unreliable measure or whether people are indeed implicitly indifferent towards free will and determinism.

Methods (page 10)

In order to assess the reliability of the IAT measure, we will compute and report internal consistencies using the modal strategy for IAT data (i.e. Spearman-Brown corrected split half correlations, see e.g. Greenwald et al. JPSP 2003).

Regarding the validity of the free will IAT, past research has shown that the IAT is a valid measure of implicit attitudes across various different domains (e.g. race, sexual orientation, Greenwald et al. JPSP 2003 & 2009, Nosek et al., ExpPsy 2007). This research demonstrated that implicit attitudes are separate from explicit attitudes, even if both were correlated. We believe it is unlikely that the free will IAT used here would lack the validity that the majority of other IATs have. We now state more explicitly in the manuscript that the IAT is a much used and validated measure.

Methods (pages 6/7)

The IAT measures automatic associations between these categories through reaction time measures, and the IAT here was administered and scored following standard IAT procedures (Greenwald et al. JPSP 2003). Past research demonstrated that the IAT is a reliable and valid measure of implicit attitudes (Greenwald et al. JPSP 2003, 2009, Nosek CurrDirPsycholSci 2007, Nosek & Smyth ExpPsychol 2007), across various domains (e.g. race, sexual orientation).

We agree with the reviewer that correlating free will IAT scores with other IAT scores would have been a useful analysis. Unfortunately, this is impossible in the current dataset, as the vast majority of participants only performed a single IAT, but future research might be able to provide such data. Yet, there are different ways in which we can assess both the convergent and divergent validity of the free will IAT. For this purpose, we will follow the analytic approach established by Nosek et al. ERSP 2007. We will perform a multiple regression, simultaneously predicting IAT scores from: gut feelings, actual feelings, and other people's feelings. If the IAT reflects implicit judgments, it should be more strongly related to gut feelings, than to deliberative evaluations, or merely preferences we are aware of in others. We thus predict the IAT to be most strongly related to gut feelings, and to a lesser degree to actual and other people's feelings. This analysis will thus provide additional evidence for the validity of the specific free will IAT used here.

Methods (pages 10/11)

In order to assess the convergent and divergent validity of the IAT, we will perform a multiple regression (following the approach described in Nosek et al. ERSP 2007), simultaneously predicting IAT scores from: gut feelings, actual feelings, and other people's feelings. These three predictors reflect self-rated intuitive reactions, deliberative reactions, and preferences we are aware of in others. All three items are rated on a 10-point scale (from "strongly negative" to "strongly positive"). If the IAT was mostly reflecting fast, intuitive evaluations, we would expect the strongest relation to gut feelings, while we would also expect relations to actual feelings and other people's feelings to be lower.

Appendix D

David Wisniewski
FWO [PEGASUS]² Marie Skłodowska-
Curie Fellow

E david.wisniewski@ugent.be
T +32 (0)9 2649428

Henri Dunantlaan 2
9000 Gent
Belgium

DATE PAGE
25 June 2020 1/1

Submission Stage 2 Registered Report

Dear RSOS team,

We concluded our pre-registered analyses for the approved Stage 1 Registered Report “Cultural pressure and biased responding in free will attitudes”, which we now submit as a Stage 2 Registered Report to *Royal Society Open Science*.

All planned analyses have been executed as pre-registered in the approved Stage 1 protocol. However, we want to point out that the original estimation of sample sizes in the Stage 1 Registered Report were not all correct. For Hypotheses 4-6, the effective sample size was slightly lower than originally anticipated. Importantly, sample sizes are still large enough to perform our planned analyses and draw conclusions from the results. Additionally, for Hypothesis 1, we originally reported that we would perform a t-test against 0, which was an error. While this t-test is correct for explicit attitudes, the t-test for implicit attitudes needs to be performed against 0.5. We now corrected this mistake and updated sample sizes and power estimations in Table 1 (page 13).

The URLs to the data, code, and stage 1 protocol can be found on page 28 of the submitted manuscript. No other data than the exploratory dataset reported in stage 1 was subjected to the pre-registered analyses prior to IPA. We would like to point out that the full data set reported here is currently still embargoed. Reviewers can get access to the full data set by contacting the data curator (link in the manuscript), and the full data set will be made freely available online in the future.

We believe that this Registered Report will have a significant impact on the literature and look forward to your feedback.

With kind regards,

David Wisniewski, Emiel Cracco, Carlos González-García, Senne Braem, Ian Hussey

Appendix E

Associate Editor:

The Stage 2 manuscript was returned to one of the original Stage 1 reviewers -- the other reviewer was unavailable. The reviewer's assessment is positive and mirrors my own (non-specialist) opinion having also read the manuscript again. The reviewer offers a thoughtful critique of the conceptual basis of the research, in large part as a reflection on future steps. I think this is a useful contribution, especially given that the journal's open review policy. However, in revising, please confine any responses to the issues raised (and consideration of limitations) to the Discussion, as it is not possible to alter or relitigate the rationale and methods approved at Stage 1. Provided the authors are able to respond thoroughly to all points raised (including provision of some additional detail -- see the reviewer's point 5), either through revision or rebuttal, a revised Stage 2 manuscript is likely to be accepted without requiring further in-depth review.

We thank the editor for the positive evaluation of our manuscript. The reviewer's comments are indeed valuable and we revised the manuscript accordingly. Except for some clarifying statements in the methods section, we limited changes to the discussion section, as requested.

Reviewer: 1

Free will beliefs have recently received a lot of attention in the literature from experimental philosophy, social psychology, and cognitive science. These authors make a novel contribution to this area of research by switching focus from people's beliefs about free will to their attitudes about free will. As such, this paper represents an important step in a new direction that broadens the scope of research on the role that free will plays in people's lives. It also opens up a fertile new area for empirical investigation. I nevertheless have several concerns about how attitudes about free will have been operationalized, which in turn problematizes the conclusions drawn from the findings.

Despite these concerns, I think the paper merits publication given the novelty of the authors' focus (and besides, the fact that they had a very large sample and didn't find any correlations between free will attitudes and a battery of personality measures is a very important finding that in itself makes an important contribution to the field). The concerns I raise below are not necessarily concerns I think the authors need to address for the purposes of publishing this paper. They are concerns that I think would need to be addressed in future work now that they have opened up this new line of investigation.

As will be clear, several of the issues I raise are based on the extant data and variables the authors used for their analyses. For a preliminary investigation, it makes perfect sense that they would use some of the variables from the IAT literature (even if they are suboptimal). Despite the problems with some of these variables, I think the authors are to be applauded for pushing research on the psychology of free will in a new and important direction. But moving forward, more care would need to be taken to improve upon the measures that are used here to explore people's attitudes about free will, determinism, and related constructs.

We thank the reviewer for the positive evaluation of our submission, and fully agree that future research will need to optimize some of the methods used here. We address each of the reviewer's points below.

First, it sounds very odd and unnatural to say, “I like free will,” or “I don’t like free will.” This isn’t how one would naturally express the fact that one values free will. The “I like...” locution makes more sense when talking about fruits, vegetables, music genres, people, etc. It sounds odd when it comes to free will. Something like “I value free will” or “Having free will is important to me” both sound much more natural. In my own case, I don’t think we have free will. But I wouldn’t say I don’t like it (even if I don’t necessarily value it). Given that this entire project is about people’s attitudes about free will, I don’t think this is a marginal issue. The items used for measuring these attitudes are crucial. And I worry that some of the items are worded in odd and unnatural ways.

We agree that saying “I like free will” seems somewhat unnatural, and we were not clear enough in the manuscript how the explicit attitude (valence) items were worded in the study. In fact, the items were “How positive or negative do you feel towards free will / determinism?”, which does not use the word “like”. We now included the exact wording of the valence items in the manuscript (page 10) and replaced “liking free will” with “valuing free will” throughout the manuscript (except the preregistered introduction section), as this better captures the concept of free will attitudes. We also point out more explicitly that future research should ensure that explicit attitude items are optimized for the free will context.

Pages 22/23:

Additional effort should be invested into the precise wording of the attitude items. The AIID project used a wide range of different explicit attitude items, and here we focused on valence items (“How positive or negative do you feel towards free will / determinism?”). For research specifically focusing on free will, it might be that other items (e.g. “Having free will is important to me.”) are better suited to capture free will attitudes.

Second, there is a related concern when it comes to the relationship between free will beliefs and free will attitudes. If I don’t believe that free will exists, then it’s hard to know what sense it makes to say that “I like free will.” It would be akin to not believing in souls or ghosts, but saying that I nevertheless like souls or ghosts. If I don’t believe that free will exists, what sense does it make for me to either like it or dislike it? Instead, it seems like the issue is whether or not I would prefer to have free will regardless of whether I actually have it. But to get at this, one would need a different approach than the one adopted by the authors. A nested or branching approach would work better. Something like: First, you ask whether they believe in free will. If yes, you ask how much they value it? If no, you ask how much it matters to [or bothers] them that they don’t have it. Or something along those lines...

We agree that the relationship between free will attitudes and beliefs is complex (and understudied). Here, we investigate attitudes only and cannot make strong claims about their relationship to beliefs based on our data. In order to characterize this relationship, future work clearly needs to study attitudes and beliefs simultaneously, which we now point out more explicitly in the discussion section.

Page 23:

Third, future work should focus on investigating free will beliefs and attitudes simultaneously, in order to describe and explain their interactions. Here, we focused on attitudes only, but in order to understand belief attitude interaction, e.g. whether free will attitudes mediate effects of free will belief manipulations on behavior, we need to assess beliefs and attitudes within the same

participants. This would also allow us to test whether free will attitudes are conditional on specific free will beliefs (e.g. "I can only value free will if I believe in it."), or vice versa.

In principle, it seems that attitudes and beliefs can be independent. There are concepts that are quite important to some people that one believes exist and likes (e.g. evolution). Others, one might believe in but dislike (e.g. global heating), or disbelieve and dislike (e.g. creationism). Lastly, there are concepts central to the self-concept of some people that they do not think exist, but like anyway (e.g. world peace). In which of these categories free will and determinism fall will likely differ between people. Thus, at this point it seems that future work should not assume specific conditional dependencies between free will beliefs and attitudes. But clearly, this is an empirical question that will have to be addressed in future work, and testing for conditional dependencies (in any direction) is a very good starting point for that.

Third, the IAT items the authors used are problematic. Free will is operationalized using "intention", "freedom", and "choice", while determinism is operationalized using "fixed", "destined", "arranged". While it's true that the exercise of free will often involves intentions, freedom, and choice, these concepts are dissociable from free will in myriad ways. For instance, someone like me who doesn't believe in free will might nevertheless believe in intentions, choices, and other kinds of freedom (e.g., political). So, it's not clear to me that these items shed light on free will per se. As for the determinism items, while fixed and destined seem good enough proxies, arranged is positively bizarre (since there are tons of uses of the term which have nothing to do with determinism). In short, I am not sure how illuminating the results of the IAT analysis are given that the terms used are problematic. I realize that the authors did not design these items, but insofar as their analysis is built on them, they inherit whatever problems are associated with them.

We agree that the items used in the IAT are problematic from a philosophical point of view. One of the problems in most of the research on free will beliefs or attitudes is the balance between conceptual precision and accessibility for the participant. On the one hand, researchers need to define their concepts as precisely as possible in order to dissociate the many subtly different philosophical positions as best they can. This might be one of the reasons why much research rests on vignette or questionnaire designs, which allow a much higher level of detail and precision than the IAT. On the other hand, experimental procedures still need to be accessible and understandable to lay participants. Conceptual precision might be a limiting factor if a degree in philosophy is needed to grasp subtle conceptual differences. In this context, the IAT does not allow for a high degree of precision, and almost necessarily needs to (over)simplify complex concepts to just a few words.

We already point out some of the difficulties with using IATs for free will attitude assessment in the discussion (page 22), and we now added an additional section specifically highlighting the use of the IAT items. We agree that alternatives to the IAT are needed in the future.

Page 22

Lastly, the words used to describe free will (intention, freedom, choice), and determinism (fixed, destined, arranged) have many uses that have little to do with the philosophical concepts of either free will or determinism, and might not precisely capture them. For instance, "arranged" is often used to describe arranging a meeting, which has little to do with physical determinism. Taken together, this suggests that future research should invest effort into developing alternative measures of implicit attitudes towards free will.

Fourth, the authors claim:

“Besides the IAT, participants were also given between 27 and 29 possible self-report evaluation items about free will and determinism, out of a pool of 79 self-report attitude items. This included items such as valence, liking, gut feelings, actual feelings, confidence, and attitude stability. Here, we will focus on assessing self-reported explicit attitudes towards free will and determinism and self-reported culture pressure to like/dislike free will or determinism (for a full description of the variables used see below).”

Perhaps I missed it, but I couldn’t find the list of these variables. They should be included somewhere in the paper.

We now include both a reference to the full list of items (<https://osf.io/atymr/>), as well as the exact wording of each item (<https://osf.io/3sg5e/>) on page 8.

Fifth, the authors claim:

“In order to assess explicit FWAs, participants rated both free will and determinism separately on a valence scale (10-point scale, from “strongly negative” to “strongly positive”).”

But they do not provide the precise wording that the participants received for these items. What were the directions? Where they asked, “What are your attitudes or feelings towards the following terms?”

The exact wording of these items was “How positive or negative do you feel towards free will / determinism?”, which is now included in the manuscript (page 10, see also point 1 above).

Sixth, I am worried that the authors have made it impossible for compatibilists to accurately report their explicit attitudes. Consider, for instance, the following:

“Responses to this item ranged from “Strongly prefer free will over determinism” to “Strongly prefer determinism over free will”.

Compatibilists don’t see any conflict between free will and determinism so it’s not clear how they are supposed to respond to this kind of forced ranking of preferences. Indeed, some compatibilists think that without determinism, we can’t be free. So, for people who have these views—and surely there are some people with these views—it’s not clear these answer choices capture their actual preferences.

We fully agree that compatibilists might have problems in accurately reporting their attitudes. We already discuss this issue for the implicit attitude measures used (page 22). In the stage 1 registered report, we decided to use difference scores for explicit attitude ratings (valence free will – valence determinism) in order to make them comparable with the IAT scores that use an analogous difference scoring. Participants separately rated the valence of both free will and determinism, and these two items were then combined into a single difference score (see page 10 for full details). This in fact imposes an incompatibilist view on participants.

For the explicit attitude items, it is possible in principle to analyze responses separately for free and determinism, and we already did so in an additional non-preregistered analysis (page 15). We show that more people value free will positively and determinism negatively, than the opposite. Nevertheless, we now state in the discussion section that explicit attitudes towards free will and determinism should be investigated separately to allow for the identification of compatibilists.

Additionally, in order to make explicit and implicit attitude measures comparable here, we computed a difference score from free will and determinism valence ratings (val_diff). While this is useful from a methodological perspective, this procedure suffers from some of the same issues as the free will IAT. Just as the IAT cannot detect people who value both free will and determinism positively, as some compatibilists might do, the val_diff items cannot do so either. In the future, explicit attitude items for free will and determinism should be investigated separately, to more easily identify compatibilist participants.